# Selecting a climate model subset to optimise key ensemble properties

Nadja Herger[1], Gab Abramowitz[1], Reto Knutti[2,3], Oliver Angélil[1], Karsten Lehmann[4], and Benjamin M. Sanderson[3]

[1]Climate Change Research Centre and ARC Centre of Excellence for Climate System Science, UNSW Sydney, Sydney NSW 2052, Australia
[2]Institute for Atmospheric and Climate Science, ETH Zurich, Zurich, Switzerland
[3]National Center for Atmospheric Research, Boulder, Colorado, USA
[4]Satalia, Berlin, Germany

*Correspondence to:* Nadja Herger (nadja.herger@student.unsw.edu.au)

**Abstract.** End-users studying impacts and risks caused by human-induced climate change are often presented with large multi-model ensembles of climate projections whose composition and size are arbitrarily determined. An efficient and versatile method that finds a subset which maintains certain key properties from the full ensemble is needed, but very little work has been done in this area. Therefore, users typically make their own somewhat subjective subset choices and commonly use the equally-weighted model mean as a best estimate. However, different climate model simulations cannot necessarily be regarded as independent estimates due to the presence of duplicated code and shared development history.

Here, we present an efficient and flexible tool that makes better use of the ensemble as a whole by finding a subset with improved mean performance compared to the multi-model mean while at the same time maintaining the spread and addressing the problem of model interdependence. Out-of-sample skill and reliability are demonstrated using model-as-truth experiments. This approach is illustrated with one set of optimisation criteria but we also highlight the flexibility of cost functions, depending on the focus of different users. The technique is useful for a range of applications that, for example, minimise present day bias to obtain an accurate ensemble mean, reduce dependence in ensemble spread, maximise future spread, ensure good performance of individual models in an ensemble, reduce the ensemble size while maintaining important ensemble characteristics, or optimize several of these at the same time. As in any calibration exercise, the final ensemble is sensitive to the metric, observational product and pre-processing steps used.

## 1 Introduction

Multi-model ensembles are an indispensable tool for future climate projection and the quantification of its uncertainty. However, due to a paucity of guidelines in this area, it is unclear how best to utilise the information from climate model ensembles consisting of multiple imperfect models with a varying number of ensemble members from each model. Heuristically, we understand that the aim is to optimise the ensemble performance and reduce the presence of duplicated information. For such an optimisation approach to be successful, metrics that quantify performance and duplication have to be defined. While there are examples of attempts to do this (see below), there is little understanding of the sensitivity of the result of optimisation to the

subjective choices a researcher needs to make when optimising.

As an example, the equally-weighted multi-model mean (MMM) is most often used as a "best" estimate for variable averages (Knutti et al., 2010), as evidenced by its ubiquity in the Fifth Assessment Report of the United Nations Intergovernmental Panel on Climate Change (IPCC, 2014). In most cases, the MMM – which can be regarded as an estimate of the forced climate response – performs better than individual simulations. It has increased skill, consistency and reliability (Reichler and Kim, 2008; Gleckler et al., 2008) as errors tend to cancel (Knutti et al., 2010), although part of that effect is the simple geometric argument of averaging (Annan and Hargreaves, 2011). However, model democracy ("one model, one vote") (Knutti, 2010) does not come without limitations. A lack of independence in contributions to the Coupled Model Intercomparison Project Phase 5 (CMIP5) (Taylor et al., 2012) archive (Masson and Knutti, 2011; Knutti et al., 2013), where research organisations simply submit as many simulations as they are able to (thus often referred to as "ensemble of opportunity" (Tebaldi and Knutti, 2007)), means that it is extremely unlikely that the MMM is in any way optimal. Different research groups are known to share sections of code (Pincus et al., 2008), literature, parametrizations in their models, or even whole model components, so that at least heuristically, we understand that individual model runs do not necessarily represent independent projection estimates (Abramowitz, 2010; Abramowitz and Bishop, 2015; Sanderson et al., 2015a). Ignoring the dependence of models might lead to a false model consensus, poor accuracy and poor estimation of uncertainty.

Instead of accounting for this dependence problem, most studies use whatever models and ensembles they can get and solely focus on selecting ensemble members with high individual performance (e.g., Grose et al. (2014)). They assume that if individual members of an ensemble perform well, then the mean of this ensemble will also have high skill. As we demonstrate later, this is not always the case, and can potentially be highly problematic.

Given that climate models developed within a research group are prone to share code and structural similarities, having more than one of those models in an ensemble will likely lead to duplication of information. Institutional democracy as proposed by Leduc et al. (2016) can be regarded as a first proxy to obtain an independent subset. However, in this case dependence essentially reflects an *a priori* definition of dependence that may not be optimal for the particular use case (e.g., variable, region, metric, observational product). There are also a few cases in which a model is shared across institutes and thus this approach would fail (e.g., NorESM is built with key elements of CESM1), or at least need to evolve over time.

Only a few studies have been published that attempt to account for dependence in climate model ensembles. A distinction can be made between approaches that select a discrete ensemble subset and those that assign continuous weights to the ensemble members. For example, Bishop and Abramowitz (2013) proposed a technique in which climate model simulations undergo a linear transformation process to better approximate internal climate system variability, so that models and observations were samples from a common distribution. This weighting and transformation approach was based on a mean square difference adherence to an observed product over time and space within the observational period, with ensemble spread at an instant in time calibrated to estimate internal variability. The same process was also used for future projections, with the danger of

over-fitting mitigated through out-of-sample performance in model-as-truth experiments (Abramowitz and Bishop, 2015). In their approach, they solely focus on variance by looking at time series.

Another method also using continuous weights but considering climatologies rather than time series was proposed by Sanderson et al. (2015a). It is based on dimension reduction of the spatial variability of a range of climatologies of different variables. This resulted in a metric to measure the distance between models, as well as models and observational products, in a projected model space (Abramowitz et al. (2008) is another example of an attempt to do this). Knutti et al. (2017a) aims to simplify the approach by Sanderson et al. (2015a), where models which poorly agree with observations are down-weighted, as are very similar models that exist in the ensemble, based on RMSE distance. Projections of the Arctic sea ice and temperatures are provided as a case study. Perhaps not surprisingly, the effect of weighting the projections is substantial, and more pronounced on the model spread than its best estimate.

Sanderson et al. (2015b) proposes a method that finds a diverse and skillful subset of model runs that maximises inter-model distances, using a stepwise model elimination procedure. Similar to Sanderson et al. (2015a), this is done based on uniqueness and model quality weights.

Sanderson et al. (2017) applied a similar continuous weighting scheme to climatological mean state variables and weather extremes in order to constrain climate model projections. Only a moderate influence of model skill and uniqueness weighting on the projected temperature and precipitation changes over North America was found. As under-dispersion of projected future climate is undesirable, only a small reduction in uncertainty was achieved.

In the previous paragraph we discussed approaches that assign continuous weights to model runs. Regional dynamical down-scaling presents a slightly different problem to the one stated above, as the goal of regional climate models is to obtain high resolution climate simulations based on lateral boundary conditions taken from global climate models (GCMs) or reanalyses (Laprise et al., 2008). One might therefore attempt to find a small subset of GCMs that reproduces certain statistical character-istics of the full ensemble. In this case the issue of dependence is critical, and binary weights are needed, since computational resources are limited. Many research groups can only afford to dynamically downscale a few GCM simulations. With *binary* we refer to the weights being either zero or one, and thus a model run is either discarded or part of the subset. Such an approach is presented in Evans et al. (2013), where independence was identified to be central for creating smaller ensembles.

The problem of defining and accounting for dependence is made more challenging by the fact that there is no uniformly agreed definition of dependence. A canonical statistical definition of independence, that two events A and B are considered to be independent if the occurrence of B does not affect the probability of A, P(A), so that P(A|B)=P(A). As discussed by Annan and Hargreaves (2017), there could, however, be many approaches to applying this definition to the problem of ensemble pro-jection that could potentially yield very different results. An appropriate course of action regarding what to do if two models are identified to be co-dependent does not follow directly from this usual definition of independence.

One disadvantage of many of these studies is that they are technically challenging to implement and therefore discourage frequent use. Further, the sensitivity of each approach to the choice of metrics used, variables included and uncertainties in observational products is largely unexplored. This leads to a lack of clarity and consensus on how best to calibrate an ensemble for a given purpose. Often, out-of-sample performance has not been tested, which we consider essential when looking at ensemble projections.

The aim of this study is to present a novel and flexible approach that selects an optimal subset from a larger ensemble archive in a computationally feasible way. Flexibility is introduced by an adjustable cost function which is allowing this approach to be applied to a wide range of problems. The meaning of "optimal" varies depending on the aim of the study. As an example, we will choose a subset of the CMIP5 archive that minimises regional biases in present day climatology, based on RMSE over space using a single observational product. The resulting ensemble subset will be optimal in the sense that its ensemble mean will give the lowest possible RMSE against this observational product of any possible combination of model runs in the archive. The more independent estimates we have, the more errors tend to cancel. This results in smaller biases in the present day which reduces the need for bias correction. Such an approach with binary (0/1) rather than continuous weights is desired to obtain a smaller subset that can drive regional models for impact studies, as this is otherwise a computationally expensive task. More precisely, it is the number of zero weight that leads to some models being discarded from the ensemble. Out-of-sample skill of the optimal subset mean and spread is tested using model-as-truth experiments. The distribution of projections using model runs in the optimal subset is then assessed.

We then examine the sensitivity of this type of result to choices of the cost function (by adding additional terms), variable and constraining data set. We argue that optimally selecting ensemble members for a set of criteria of known importance to a given problem is likely to lead to more robust projections for use in impact assessments, adaptation and mitigation of climate change.

This approach is not meant to replace or supersede any of the existing approaches in the literature. Just as there is no single best climate model, there is no universally best model weighting approach. Whether an approach is useful depends on the criteria that are relevant for the application in question. Only once the various ensemble selection approaches have been tailored to a specific use-case, can a fair comparison be made. Flexibility in ensemble calibration by defining an appropriate cost function that is being minimised and metric used is key for this process.

In the following section, we introduce the model data and observational products used throughout this study. Section 3 contains a description of the method used, which includes the pre-processing steps of the data and three ensemble sub-sampling strategies, one of which is the novel approach of finding an optimal ensemble. In section 4 we examine the results by first giving the most basic example of the optimisation problem. We then expand on this example by examining the sensitivity of those results to different choices of the user and highlight the method's flexibility in section 4.1. Out-of-sample skill is tested in

section 4.2.1 using model-as-truth experiments to ensure that our approach is not overfitting on the current present-day state. Once that has been ensured, we present future projections based on this novel approach (section 4.2.2). Finally, section 5 contains the discussions and conclusions.

## 2   Data

We use 81 CMIP5 model runs from 38 different models and 21 institutes which are available in the historical period (1956–2013; RCP4.5 after 2005) and RCP4.5, RCP8.5 period (2006–2100) (see Table 1 in the Supplementary Information (SI)). We examine gridded monthly surface air temperature (variable: tas) and total monthly precipitation (variable: pr). Results shown here are based on raw model data (absolute values), although repeat experiments using anomalies (by subtracting the global mean climatological value from each grid cell) were also performed (not shown here).

Multiple gridded observation products for each variable were considered with each having different regions of data availability (see Table 2 and additional results in the SI). Model and observation data were remapped using a first order conservative remapping procedure (Jones, 1999), to either 2.5° or 5° spatial resolution, depending on the resolution of the observational product (see SI Table 2). For the projections, the model data was remapped to a resolution of 2.5°. For observational products

whose data availability at any grid cell changes with time, a minimal two-dimensional mask (which does not change over time) was used. The remaining regions were masked out for both the observational product and the model output.

## 3   Method

We first illustrate the technique by considering absolute surface air temperature and total precipitation climatologies (time-means at each grid-cell), based on 1956–2013. The land-only observational product CRUTS, version 3.23 (Harris et al., 2014)

is used for both variables and model data is remapped to the same spatial resolution and masked based on data availability in this product.

Next, we select an ensemble subset of size K from the complete pool of 81 CMIP5 simulations, using three different approaches:

**Random ensemble**: As the name implies, the random selection consists of randomly selected model runs from the pool of 81 without repetition. This procedure is repeated 100 times for each ensemble size in order to gauge sampling uncertainty. The uncertainty range was found not to be very sensitive to the number of iterations.

**Performance ranking ensemble**: This ensemble consists of the "best" performing model runs from the ensemble in terms of their RMSE (based on climatology — time means at each grid-cell). Individual model runs are then ranked according to their

performance and only the best K model runs are chosen to be part of the subset.

**Optimal ensemble**: In this case we find the ensemble subset whose mean minimises RMSE, out of all possible K-member subsets. This is non-trivial – there are $2.12 \cdot 10^{23}$ possible ensembles of size 40, for example, so that a "brute-force" approach is simply not possible. Instead, we use a state-of-the-art mathematical programming solver (Gurobi (2015)). It minimises the MSE between the mean of K model runs and the given observational product, by selecting the appropriate K model runs. Hereinafter we refer to the ensembles (one obtained for each K) derived from this approach as "optimal ensembles" and the optimal ensemble with the overall lowest RMSE as the "optimal subset". Note, that optimal refers to the specific question at hand that the ensemble is calibrated to. The ensemble would no longer be optimal if the specific application changes. The problem itself is a mixed integer quadratic programming problem because the decisions are binary (that is: model run is in the set or not), the cost function is quadratic (see Eq. (1)), and the constraint is linear. Such a problem is solved using a branch-and-cut algorithm (Mitchell, 2002).

In the following section, we compare these three subsampling strategies with the benchmark, which is the simple unweighted multi-model mean (MMM) of all 81 runs. We then examine the sensitivity of results to the observational product, the cost function (to demonstrate flexibility by optimising more than just the ensemble mean) and other experimental choices.

## 4  Results

Figure 1 displays the area-weighted root mean square error (RMSE) of the subset mean and RMSE improvement relative to the MMM of all 81 model runs (solid horizontal line) as a function of ensemble size for the three different methods used to select subsets. The RMSE is calculated based on the climatological fields of pre-processed model output and observations. Results based on CRUTS3.23 as the observational product are shown for both surface air temperature (a) and precipitation (b). We focus on panel (a) for now. Each marker represents the RMSE of an ensemble mean, except for ensemble size one, which refers to the single best performing model run in terms of RMSE. Blue markers are used for the random ensemble, with the error bar indicating the 90% confidence interval (from 100 repetitions). The performance ranking ensemble is shown in green. For ensemble sizes one to four, the RMSE of the performance ranking ensemble increases. This is because multiple initial condition ensemble members of the same model (MPI-ESM) are ranked high, and averaging across those leads to higher dependence within the subset and thus less effective cancelling out of regional biases. Interestingly, the performance-based ensemble sometimes even performs worse than the mean of the random ensemble, which can be observed across multiple observational products and across the two variables (see SI). This is a clear example of the potential cost of ignoring the dependence between ensemble simulations. Selecting skillful but similar simulations can actively degrade the present-day climatology of the ensemble mean.

For the optimal ensemble (black circles), RMSE is initially large, the value representative of the single best performing model run (black dot being behind the green one). The RMSE of the ensemble mean rapidly decreases when more model runs are included until it reaches a minimum (red circle indicates the optimal subset over all possible ensemble sizes). That is, the RMSE improvement relative to the MMM (solid horizontal line) is largest at this ensemble size. One could investigate defining the effective number of independent models for a given application based on the optimal ensemble size (Annan and Hargreaves, 2011; Bishop and Abramowitz, 2013; Sanderson et al., 2015b; Jun et al., 2008; Pennell and Reichler, 2011), but we have not explored this idea in any detail. As more model runs are included in the ensemble, the RMSE increases again. This is expected as worse performing and more dependent model runs are forced to be included. The MMM generally outperforms every individual model run (green, black and blue dots at subset size one being above the solid horizontal line). The optimal ensemble curve in the vicinity of the lowest RMSE is often rather flat, so different ensembles with similarly low RMSE could be chosen instead if, for example, a given model is required to be part of the subset. A flat curve is also of advantage in the case when computational resources are limited and thus a small ensemble size has to be chosen (for example when global model boundary conditions are being chosen for a downscaling experiment). Here, however, we always consider the ensemble with the overall smallest RMSE (red circle) as our optimal subset even if ensembles of similar sizes are not much worse. We will discuss the black triangle markers and other horizontal lines in a later section.

Note, that as the selection of one ensemble member depends on the remaining members in the ensemble, the optimal subset is sensitive to the original set of model runs that we start out with. So, if members are added to or removed from the original ensemble, then the optimal subset is likely going to change. Any subset selection approach that does not make use of the available information about the original ensemble is most likely not optimal.

Another characteristic of the optimal ensemble is that there is not necessarily any ensemble member consistency (with increasing subset size). There are other methods which do maintain this consistency (e.g. Sanderson et al. (2015b)), but such an ensemble is no longer optimal from an ensemble mean point of view.

Of the three sub-sampling approaches, it is evident that the optimal ensemble mean is the best performing one for all ensemble sizes if the bias of the model subset average should be minimized – essentially indicating that the solver is working as anticipated. Regional biases in different models cancel out most effectively using this approach. Across different observational products, we observe an improvement in RMSE relative to the MMM of between 10–20% for surface air temperature, and around 12% for total precipitation (see Figure S1 and S2). The size of the optimal subset is significantly smaller than the total number of model runs considered in this study (see red text in Figure 1). For surface air temperature we obtain an optimal ensemble range between six to ten members and for precipitation around twelve members. This suggests that many model runs in the archive are very similar.

We achieve similar RMSE improvement if we exclude closely related model runs *a priori* and start off with a more independent set of model runs (one model run per institute), see Figure S3.

Figure 1 solely looks at the performance of the ensemble mean. A characterisation of the relationship between model simulation similarity and performance in these ensembles is shown in Figure 2. Simulation performance (in terms of RMSE) is plotted against the simulation dependence (expressed as average pairwise error correlation across all possible model pairs in the ensemble) for the three sampling techniques (3 colors). As before, CRUTS3.23 was used as the observational product, but this figure looks very similar across different variables and observational products. Circular markers are used for the average of individual members of the subset ensemble of any given size and diamond markers are used for ensemble mean. The darker the color, the larger the ensemble size. Members of the optimal ensemble (black markers) are more independent than members of other ensembles, at least in terms of pairwise error correlation. Members of the performance-ranking ensemble (green markers) however show high error correlations as closely related model runs are likely to be part of the ensemble. We thus conclude that the optimal ensemble has favourable properties in terms of low ensemble mean RMSE and low pairwise error correlation of their members. We will therefore focus on this the ability of this sampling technique for the remainder of the paper.

## 4.1  Sensitivity of results

We now develop this optimisation example to highlight the flexibility of the method. In doing so, it should become clear that calibration for performance and dependence is necessarily problem dependent. A graphical representation of the experimental choices we explore is shown in Figure 3. We explore different aspects of this flowchart below.

**Choice of observational product.** The ensembles in the previous subsection were calibrated on a single observational product (depicted in green in Figure 3). Observational uncertainty can be quite large depending on the variable and can thus result in a different optimal subset. Figure 1 for different observational products (and varying observational data availability) can be found in the supplementary material (Figure S1 and S2). HadCRUT4 for example is the same as CRUTEM4 over land, but additionally has data over the ocean. The optimal subsets derived from calibrating on those two observational products separately are quite different, which highlights the sensitivity of the calibration exercise to the chosen spatial domain. This is particularly important for impact assessments and regional climate modelling, where ensemble selection is done based on a specific region. Moreover, observational uncertainty within one observational product (instead of across the products) should also be considered to test the stability of the optimal subset. This has not been done here, but could certainly be investigated in future studies. Lastly, if multiple observational products per variable are available and all equally credible, finding a subset that is optimal using all of them is also a possibility. This could be done by putting multiple observational products into a single cost function. However, when using ensembles for inference, a lot can be learned from the spread across observational products. This additional uncertainty added by observations is ignored if all the products are combined in a single cost function. Here, we only optimise our ensemble to one observational product at a time and investigate how sensitive the optimal subset is to that choice.

**Variable choice.** The selection of the variable has a profound influence on the resulting optimal subset. This was already briefly highlighted in Figure 1, where the optimal subsets for surface air temperature (a) and total precipitation (b) consist of rather

different ensemble members. Generally, the optimal ensemble size for precipitation tends to be larger. Similar to the discussion above for the sensitivity to observational products, one might consider optimising the subset across multiple variables. This is particularly important if physical consistency across variables needs to be ensured. This could most simply be done using a single cost function that consists of a sum of standardised terms for different variables. This is similar to what has been done in Sanderson et al. (2015a) (see their Figure 1). However, this might conceal the fact that the optimal subsets for the individual variables potentially look very different. One might calibrate the ensemble on multiple variables using a Pareto solution set, similar to what has been done in Gupta et al. (1998) for hydrological models and Gupta et al. (1999) for land surface schemes. An important characteristic of such a problem is that it does not have a unique solution, as there is a trade-off between the different and non-commensurate variables. When improving the subset for one variable (i.e., RMSE reduced), we observe a deterioration of the subset calibrated on the other variable.

The presented approach can obtain an optimal subset for any given variable, as long as it is available across all model runs and trustworthy observational products exist. One might even consider using process-oriented diagnostics to give us more confidence of selecting a subset for the right physical reasons.

**Absolute values vs. anomalies.** Results presented in this study are all based on absolute values rather than anomalies. Whether or not bias-correction is required depends on the variable and the aim of the study. To study the Arctic sea ice extent for example, absolute values are a natural choice as there is a clear threshold for near ice-free September conditions. An example of where bias-correction is necessary is in the field of extreme weather. For example, mean biases between datasets must be removed before exceedance probabilities beyond some extreme reference anomaly can be calculated.

**Alternatives to climatology.** As part of the data pre-processing step, we computed climatologies (time-means at each grid cell) for the model output and observational dataset. In addition to climatologies, we decided to consider time-varying diagnostics ("trend" and "space+time"), which potentially contain information relevant for projections, which is not captured by time-means. For the "trend" diagnostic, we compute a linear trend of the corresponding variable at each grid cell and end up with a two dimensional array for each simulation and observational dataset. As a second time-varying diagnostic, we compute 10-year running means at each grid-cell to obtain a three dimensional array which is subsequently used for the analysis. This is hereafter referred to as "space+time". Subsection 4.2.1 shows (based on a model-as-truth experiment) how sensitive the ensemble can be to the diagnostics (mean, trend, or variability) chosen in the pre-processing step.

**Defining the benchmark.** To assess whether our optimal subset has improved skill, we need to define a benchmark. In Figure 1, we used the MMM of 81 model runs as our benchmark (solid line). However, other benchmarks could be used. The three horizontal lines in Figure 1 refer to three different baselines that could be used to compare against subset performance. The solid line is the MMM of all 81 model runs. We would consider this bad practice as we arbitrarily give more weight to the models represented by the largest number of members. For the dashed line, we first averaged across the ensemble members from each climate model and then average across all 38 models (same is done for the maps in Section 4.2.2). The dotted line is the ensemble mean when only allowing one run per institute to be part of the ensemble. Interestingly, the dotted line is

very often the highest one and the solid line has the lowest RMSE. One likely explanation is that the original CMIP5 archive is indirectly already slightly weighted due to a higher replication of well-performing models (Sanderson et al., 2015b). By eliminating those duplicates, our ensemble mean gets worse because regional biases do not cancel out as effectively. For the model-as-truth experiment described in subsection 4.2.1, our benchmark was also obtained by selecting one model run per institute.

**Sensitivity to the underlying cost function.** An essential part of the optimisation problem is the cost function. Comparison of all the sensitivities mentioned above is made possible only because our subsets are truly optimal with respect to the prescribed cost function. For the results above the cost function $f(x)$ being minimised by the Gurobi solver was:

$$f(x) = f_1(x) = MSE\left(\left(\frac{1}{|x|}\sum_{i\in x} m_i\right), y\right). \tag{1}$$

Here, $x$ denotes the optimal subset (with $|x|$ being the subset size), $y$ is the pre-processed observational product and $m_i$ is model simulation $i$. $MSE$ stands for the area-weighted mean squared error function.

Reasons to use ensembles of climate models are manifold, which goes hand in hand with the need for an ensemble selection approach with an adjustable cost function. Note, that we do not consider the MSE of the ensemble mean as the only appropriate optimisation target for all applications. Even though it has been shown that the multi-model average of present day climate is closer to the observations than any of the individual model runs (e.g., Gleckler et al. (2008); Reichler and Kim (2008); Pierce et al. (2009)), it has also been shown that its variance is significantly reduced relative to observations (e.g., Knutti et al. (2010)). Also, solely focusing on the ensemble mean could potentially lead to poorer performing individual models as part of the optimal subset despite getting the mean closer to observations. Errors are expected to cancel out in the multi-model average if they are random or not correlated across models. Finding a subset whose mean cancels out those errors most effectively is therefore a good proxy for finding an independent subset, at least with respect to this metric, and is sufficient as a proof of concept for this novel approach.

Of course this cost function can and should be adjusted depending on the aim of the study, as long as the expressions are either linear or quadratic. To illustrate this idea, we add two new terms to the cost function above that account for different aspects of model interdependence:

$$f(x) = \frac{f_1(x) - \mu_1}{\sigma_1} + \frac{f_2(x) - \mu_2}{\sigma_2} - \frac{f_3(x) - \mu_3}{\sigma_3} \tag{2}$$

Here, minimising $f(x)$ will involve minimising the first and second terms in Equation (2) and maximising the third term (note the minus sign in front of term 3). To ensure that the three terms all have a similar magnitude and variability, we subtract the mean ($\mu$) and divide by the standard deviation ($\sigma$) derived from 100 random ensembles of a given ensemble size. The function $f_1(x)$ is the same as in Eq. (1). It minimises the MSE between the subset mean of a given size and the observational

product $y$. The second and third terms can be written as follows:

$$f_2(x) = \frac{1}{|x|} \sum_{i \in x} MSE(m_i, y) \tag{3}$$

$$f_3(x) = \frac{2}{|x| \cdot (|x| - 1)} \sum_{i \neq j \in x} \frac{MSE(m_i, m_j)}{\frac{1}{2} \Big( MSE(m_i, y) + MSE(m_j, y) \Big)} \tag{4}$$

The function $f_2(x)$ in the second term ensures that the mean MSE between each ensemble member and the observational product is minimised. So, this term is related to the performance of the individual ensemble members — we want to avoid very poorly performing members being in the final ensemble. It would of course also be possible to make an a priori decision on which models should be considered before starting the optimisation process. The function $f_3(x)$ averages the pairwise MSE distances between all ensemble members and then divides by the mean performance. This should be maximised and helps to

avoid clustering by ensuring that the ensemble members are not too close to each other relative to their distance to the observational product. This is a way to address dependence in ensemble spread. It also makes it harder for the algorithm to select multiple members from the same model. Sanderson et al. (2017) used a similar idea of calculating pairwise area-weighted root mean square differences over the spatial domain to obtain an inter-model distance matrix. This matrix is then normalised by the mean inter-model distance to obtain independence weights as a measure of model similarity.

Based on the climatological metric, Gurobi can solve Eq. (2) within a few seconds for any given subset size. Finding an optimal solution without this solver would have been impossible within a reasonable amount of time. Results show that the RMSE of the optimal ensemble mean based on Eq. (2) is almost as low as for Eq. (1), see Figure 1 (black circles for Eq. (1) and triangles for Eq. (2)). However, the individual members of the optimal ensemble based on the cost function with three terms

seem to have a better average performance and are slightly more independent. This might be of advantage if end users want to avoid having multiple ensemble members from the same model in the optimal subset. Term 3 in Eq. (2) will take care of that. Moreover, term 2 will make sure that bad performing model runs are excluded from the optimal subset. In other words, explicitly considering single model performance and eliminating obvious duplicates does not significantly penalize the ensemble mean performance. The magnitude of the three terms in eq. (2) as a function of the ensemble size are shown in Figures S8 and

S9.

The cost function presented in this study solely uses MSE as a performance metric. There are of course many more metrics available (e.g., Xu et al. (2016); Taylor et al. (2001); Gleckler et al. (2008); Baker and Taylor (2016)) that we might choose to implement in this system for different applications. So as not to confuse this choice with the workings of the ensemble selection

approach, however, we illustrate it with RMSE alone, as this is what most existing approaches in this field use to define their performance weights (e.g., Knutti et al. (2017a); Sanderson et al. (2017); Abramowitz and Bishop (2015)).

For those concerned about overconfidence of the ensemble projections (due to the "unknown unknowns"), one could add another term which maximises future spread. This would result in an ensemble which allows to explore the full range of model responses. It is also possible to start weighting the terms of the cost function differently depending on what seems more important.

## 4.2 Application to the future

### 4.2.1 Testing out-of-sample skill

The optimal selection approach is clearly successful at cancelling out regional biases in the historical period, where observations are available. We refer to this period as "in-sample". Is a model that correctly simulates the present-day climatology automatically a good model for future climate projections? To answer this question, we need to investigate if regional biases persist into the future, and determine whether the approach is fitting short term variability. In other words, we have to ensure that our subset-selection approach is not overfitting on the available data in-sample which can potentially lead to spurious results out-of-sample. This is done by conducting model-as-truth experiments. This should give an indication of whether subselecting in this way is likely to improve future predictability or if we are likely to be overconfident with our subset. Rigid model tuning for example could cause the ensemble to be heavily calibrated on the present-day state. An optimal subset derived from such an ensemble would not necessarily be skillful for future climate prediction as we are dealing with overfitting and we are not calibrating to biases that persist into the future. This is exactly where model-as-truth experiments come into play. For this purpose, one simulation per institute is considered to be the "truth" as though it were observations, and then the optimal subset from the remaining 20 runs (one-per-institute) is determined for the in-sample period (1956–2013), based on the cost function in Eq. (1). The optimal ensemble's ability can then be tested in the out-of-sample 21st century, since we now have "observations" for this period. Results are then collated over all possible simulations playing the role of the "truth". In all our model-as-truth experiments, near relatives were excluded as truth, because members from the same model are likely to be much closer to each other than to the real observational product. This subscription to institutional democracy is consistent with what was found by Leduc et al. (2016) to prevent overconfidence in climate change projections. Sanderson et al. (2017) also removed immediate neighbours of the truth model from the perfect model test when deriving the parameters for their weighting scheme.

Figure 4 shows the results of the model-as-truth experiment for surface air temperature for the climatological field, the linear trend and space + time, as described in Section 4.1 (paragraph: "Alternatives to climatology"). Panel (a) shows global absolute mean temperature time series for the in- and out-of-sample periods. The in-sample period, in which the optimal subset is found for each model as truth is 1956–2013. For the climatological metric and the space + time metric, the same subset was tested out-of-sample in 2071–2100 using the same truth as in the in-sample period. The out-of-sample period for the trend metric is 2006–2100, as 30 years are not long enough to calculate a linear trend at each grid-cell without internal variability potentially

playing a big role. Both in- and out-of-sample data undergo the same pre-processing steps. The mask which was used for those calculations is shown in the lower right corner of panel (a).

Figures 4b–d show the RMSE improvement of the optimal subset for a given size relative to the mean of all remaining 20 simulations for each simulation as truth. The black curve is the in-sample improvement and the blue curve is the out-of-sample

improvement for RCP8.5 averaged across all truths. The shading represents the spread around the mean. Results for RCP4.5 look very similar and are therefore not shown here.

It is evident that both the climatological metric and the space + time metric have improved skill out-of-sample compared to simply taking the mean of all 20 runs. We observe an RMSE improvement almost as big as the in-sample improvement, in which we conducted the optimisation. This primarily shows the persistence of the climatological bias. Climate models which

are biased high (in terms of temperature for example) in the present day, are often at the higher end of the distribution in the projections. This is related to climate sensitivity and our approach is able to make use of this persistent bias.

The trend metric is different, however. To be clear, here we obtain the optimal subset based on a two-dimensional field with linear (58-year) trends at each grid-cell in the in-sample period. We then use this subset trained on trend values to predict the out-of-sample trend field (using the same simulation as "truth" as in the in-sample period). The RMSE improvement presented

in panel (d) is calculated from the "true" RCP8.5 trend field and the predicted trend derived from the optimal subset. We see a large in-sample improvement, but out-of-sample this skill quickly disappears. We thus conclude that the magnitude and nature of trends within individual models do not persist into the future and a subset based on this metric will not have any improved skill out-of-sample. Figure S5 shows the very weak correlation between in- and out-of-sample trend very clearly. This high-lights the difficulty of finding an appropriate metric which constrains future projections. Results for precipitation can be found

in the SI (Figure S6).

Figure 4 shed light on the increased skill of the optimal ensemble compared to the simple MMM, at least for the mean signal. We have not yet investigated the spread of the ensemble, which is as least as important, especially for impact and risk related fields. As an example, the potential danger of having a too narrow ensemble spread (overconfident projections) by neglecting

important uncertainties is highlighted in Keller and Nicholas (2015).

Results for the ensemble spread are shown in Figure 5 for surface air temperature. Panel (a) explains how the spread of the ensemble is quantified. We calculate how often the truth lies within the 10th to 90th percentile of the optimal ensemble for a given ensemble size. We derive the percentiles from a normal distribution, whose mean and standard deviation were calculated from the optimal ensemble (for a given truth and ensemble size) during the in-sample, or training period. This is done for every

grid cell and each model as truth. The curves shown in Figures 5b–d are the average of the fractions of "truth" values that lie within this range, across all grid cells and truths plotted against the subset size for the climatological field (b), the space + time (c) and linear trend (d). We would expect the truth to lie within the 10th to 90th percentile of the ensemble at least 80% of the time to avoid overconfidence. Black is used for the in-sample fraction and the two shades of blue for RCP4.5 (light blue) and RCP8.5 (dark blue). The fraction for an ensemble consisting of all 20 model runs — the benchmark in this case —

is shown with a horizontal line. The ensembles obtained based on the climatological metric and the space + time metric are

slightly over-dispersive both in- and out-of-sample, which suggests the optimal ensemble should not result in overconfidence in ensemble spread, relative to the entire ensemble. An ensemble that is overconfident can lead to projections whose uncertainty range is too narrow and thus misleading. This is the case for the trend metric, at least for smaller ensemble sizes.

Such a model-as-truth experiment can also assist with the choice of an optimal subset size for the application to projections. It does not necessarily have to be the same as the in-sample ensemble size, as aspects like mean skill improvement and reduction of the risk of underdispersion have to be considered.

Can a subset calibrated on absolute historical temperature constrain temperature *changes* in the future, as opposed to just minimising bias in the ensemble mean? This anomaly skill in the out-of-sample test is depicted in Figure 6. The setup is similar to Figure 4, but here we are predicting regional temperature change from mean values in 2006–2035 to those in 2071–2100. The optimal subset is still derived using either the climatological (b), space + time (c), or trend diagnostic (d). The only aspect that has changed is what is being predicted is now out-of-sample. The curves are the RMSE improvement relative to the MMM of 20 model runs averaged across all truths for RCP4.5 (light blue) and RCP8.5 (dark blue). Shading indicates the spread (one standard deviation) across the different truths. Results for regional precipitation change are shown in Figure S7. Panel (b) shows that there is very little to be gained by constraining the climatology in terms of out-of-sample skill. Across all metrics and variables, the subsets show hardly any RMSE improvement compared to the MMM of the 20 model runs, which is consistent with Sanderson et al. (2017). They found only small changes in projected climate change in the US when weighting models with performance on present day mean climate, and it is consistent with the fact that our field has not managed to significantly reduce uncertainties in both transient (Knutti and Sedláček, 2013) and equilibrium warming (Knutti et al., 2017b). Those findings are also consistent with Knutti et al. (2010), who found that there is only a weak relationship between model skill in simulating present-day climate conditions and the magnitude of predicted change. So, a skillful subset under present-day conditions does not guarantee more confidence in future projections. But even if the uncertainties in future projections are not strongly reduced, there is a clear advantage in reducing biases in the present day climate when using those models to drive impact models, as it reduces the need for complex bias correction methods. Ultimately, when models improve further, and the observed trends get stronger, we would expect that such methods do improve the skill of projections.

This result is partly about the discrepancy between the metric used to derive the optimal ensemble and that used to evaluate it, and reinforces how sensitive this type of calibration exercise is to the somewhat subjective choices faced by a researcher trying to post-processes climate projections. It is an important limitation that should be kept in mind when using this sampling strategy to constrain future projections.

### 4.2.2 Projections

In earlier sections we presented results based on a single observational product per variable. However, the importance of the choice of product should not be neglected. The influence of obtaining an optimal subset based on different observational products can be visualised with maps. To create Figure 7, the temperature change between the 2081–2100 and 1986–2005

climatologies was calculated for the RCP8.5 scenario using all 81 model runs, by first averaging across initial condition members before averaging the 38 models. Then, the temperature change of the optimal subset (based on a given observational product), calculated in the same way, was subtracted. The result is a map that shows the difference the optimal sampling makes to projected temperature changes. Maps are shown for the optimal subsets derived from different observations, with grey contours highlighting the area used to derive the subset. The number in brackets refers to the size of the optimal subset. Despite the maps looking quite different, we can identify some regions with consistent changes. The Southern Ocean is consistently warmer in the optimal subset and the Arctic is colder than the MMM (except for BEST, global). Generally, the optimal subset results in a cooler land surface.

Figure 8 shows the same as 7 but for precipitation change based on three different observational products. They all show an increase in precipitation in the equatorial Pacific and the western Indian Ocean and a decrease over South America.

## 5   Discussion and conclusions

We presented a method that selects a CMIP5 model subset which minimises a given cost function in a computationally efficient way. Such a calibrated smaller ensemble has important advantages compared to the full ensemble of opportunity, in particular reduced computational cost when driving regional models, smaller biases in the present day which reduce the need for bias correction, reduced dependence between the members and sufficient spread in projections. The cost function can be varied depending on the application. The simplest cost function presented here simply minimises biases of the ensemble mean. We have shown that this method accounts to some degree for the model dependence in the ensemble by the way it optimizes the ensemble mean, but closely related models or even initial condition ensemble models of the same models are not penalized and can still occur. This optimal subset performs significantly better than a random ensemble or an ensemble that is solely based on performance. The performance ranking ensemble sometimes even performs worse than the random ensemble in its mean, even though of course the individual models perform better. Depending on the application, one of the other will matter more.

We also illustrated the expansion of the cost function to optimise additional criteria, enabling an optimal subset that minimises the ensemble mean bias, the individual model biases, and the clustering of the members, or any combination thereof. One could also, for example, add a term that maximises the ensemble projection spread to avoid overconfidence. The choice of what is constrained by the cost function clearly depends on the aim of the study (e.g., present day bias, dependence issue, future spread). We highlight the importance of testing the sensitivity to the metric and observational product (incl. varying data availability) used, as they can lead to quite different results.

The lasso regression analysis method (Tibshirani, 2011) often used in the field of machine learning tries to select a subset of features (in our case: model simulations) to improve prediction accuracy. It is similar to the presented approach in a way that it also selects a subset of models by applying weights of zero. However, contrary to the method presented here, it is to our

knowledge not possible to customise the cost function that is being minimised (by default: RMSE).

Model-as-truth experiments were used to investigate the potential for overconfidence, estimate the ensemble spread, and test the robustness of emergent constraints. Based on those experiments we learned that absolute present day values constrain absolute values in the future (due to a persistent bias). However, absolute present day values do not constrain projected changes relative to a present day state.

There were other pertinent questions we did not address, of course. These include the question of how best to create an optimal subset across multiple variables and gridded observational products. This seems especially important if physical consistency across variables should be maintained. Having a pareto set of ensembles (by optimising each variable separately) rather than a single optimal subset is a potential solution, but is clearly more difficult to work with.

Using model-as-truth experiments, we observed that the skill of the optimal subset relative to the unweighted ensemble mean decreases the further out-of-sample we were testing it. This breakdown of predictability is not unexpected as the climate system reached a state it has never experienced before. This is certainly an interesting aspect which should be investigated in more depth in a future study.

Many of the points raised here are also clearly not restricted to GCMs. The same holds for regional climate models, hydrological models or perhaps ecological models. We encourage others to apply the same approach to different kinds of physically based models.

Critically, we wish to reinforce that accounting for dependence is essentially a calibration exercise, whether through continuous or discrete weights, as was the case here. Depending on the cost function, the data pre-processing and the observational product one can end up with a differently calibrated ensemble. Depending on the application, bias-correction of the model output might be appropriate before executing the calibration exercise. We suggest that the approach introduced in this study is an effective and flexible way to obtain an optimal ensemble for a given specified use case.

Future research will help to provide confidence in this method and enable researchers to go beyond model democracy or arbitrary weighting. This is especially important as replication and the use of very large initial condition ensembles will likely become a larger problem in the future global ensemble creation exercises. An approach that attempts to reduce regional biases (and therefore indirectly dependence) offers a more plausible and justifiable projection tool than an approach that simply includes all available ensemble members.

## 6 Code availability

A simplified and easily-adjustable Python code (based on the Gurobi interface) is accessible on a GitHub repository (https://github.com/nherger/EnsembleSelection/blob/master/Gurobi_MIQP_random.py). Gurobi is available via a free academic license.

## 7 Data availability

CMIP5 data can be obtained from http://cmip-pcmdi.llnl.gov/cmip5/.

*Author contributions.* N. Herger conducted the analysis, produced the figures and prepared the manuscript. G. Abramowitz came up with the core idea of ensemble selection to minimise regional biases, discussed results and helped writing the manuscript. R. Knutti contributed to discussions on the methodology and results and helped writing the manuscript. O. Angélil helped shape the methodology and contributed to the interpretation of results. K. Lehmann provided support while writing the Python code for the mathematical solver Gurobi. B. Sanderson provided useful discussions and feedback which helped shape this work.

*Competing interests.* The authors declare that they have no conflict of interest.

*Acknowledgements.* We would like to thank Jan Sedláček for providing access to the next generation CMIP5 archive based at ETHZ. We are also grateful to Steve Sherwood and Ruth Lorenz for interesting discussions which helped shape this study.

We acknowledge the support of the Australian Research Council Centre of Excellence for Climate System Science (CE110001028).

The authors acknowledge the support from the H2020 project CRESCENDO "Coordinated Research in Earth Systems and Climate: Experiments, kNowledge, Dissemination and Outreach", which received funding from the European Union's Horizon 2020 research and innovation programme under grant agreement no. 641816.

We acknowledge the World Climate Research Programme's Working Group on Coupled Modelling, which is responsible for CMIP, and we thank the climate modeling groups (listed in Table S1 in the SI) for producing and making available their model output. For CMIP the U.S. Department of Energy's Program for Climate Model Diagnosis and Intercomparison provides coordinating support and led development of software infrastructure in partnership with the Global Organization for Earth System Science Portals.

**a** - Surface Air Temperature

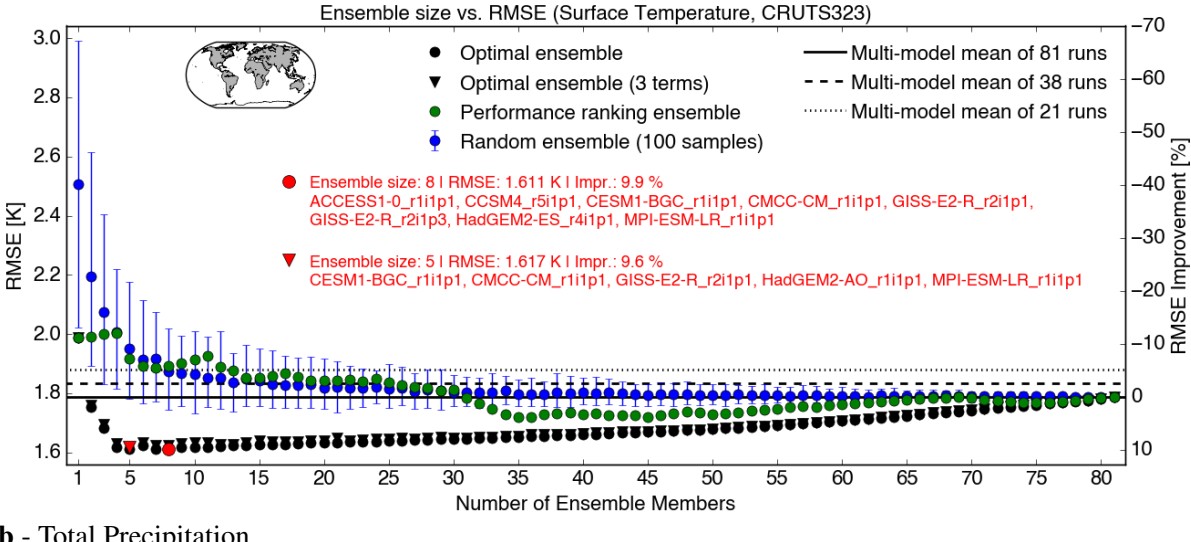

**b** - Total Precipitation

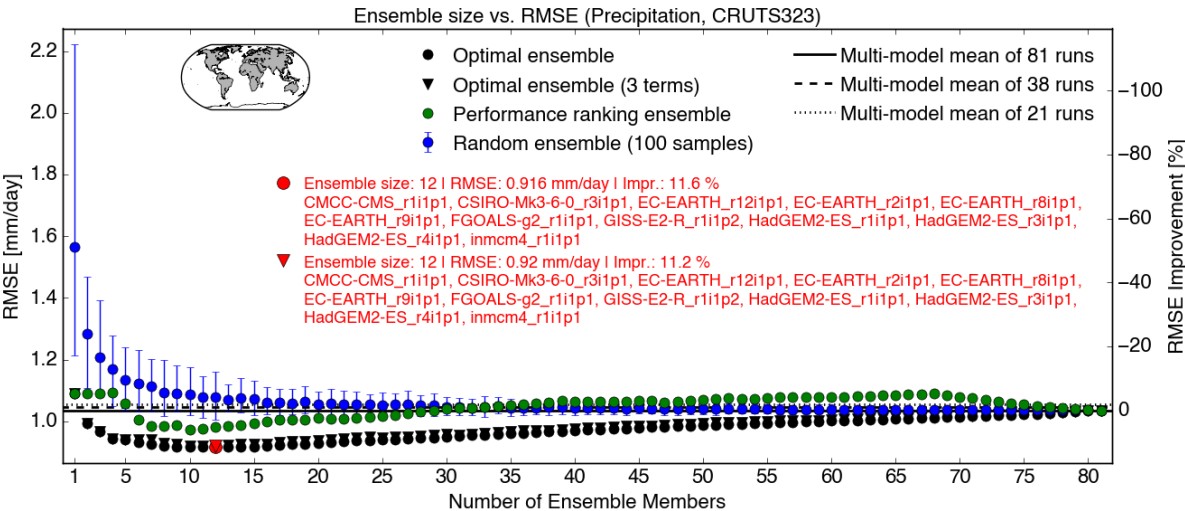

**Figure 1.** Size of the CMIP5 subset on the horizontal axis and the resulting RMSE of the ensemble mean and its improvement relative to the multi-model mean (MMM) on the vertical axes, for surface air temperature (**a**), total precipitation (**b**) and three different types of ensembles. The RMSE was calculated based on the 1956–2013 climatology of the ensemble mean and the observational product CRUTS3.23. Black dots indicate the values for the optimal ensemble, green dots the ensemble based on performance-ranking of individual members and randomly selected ensembles in blue. For the random ensemble, the dot represents the mean of 100 samples and the error bar is the 90% confidence interval. The red circle indicates the optimal subset size with the overall smallest RMSE compared to the observational product. The model simulations which are part of this optimal subset are listed in red font next to the circle. The black triangles represent the optimal ensembles for a cost function that consists of three terms (see Eq. (2)). The corresponding red triangle is the optimal subset of the black triangle cases. The map shows CRUTS3.23 coverage. The solid horizontal line indicates the RMSE value for the MMM of all 81 simulations. For the dashed line, we first averaging over the members of one model and then average over all 38 models. The RMSE of the mean of 21 simulations (1 simulation per institute) is represented with the dotted line.

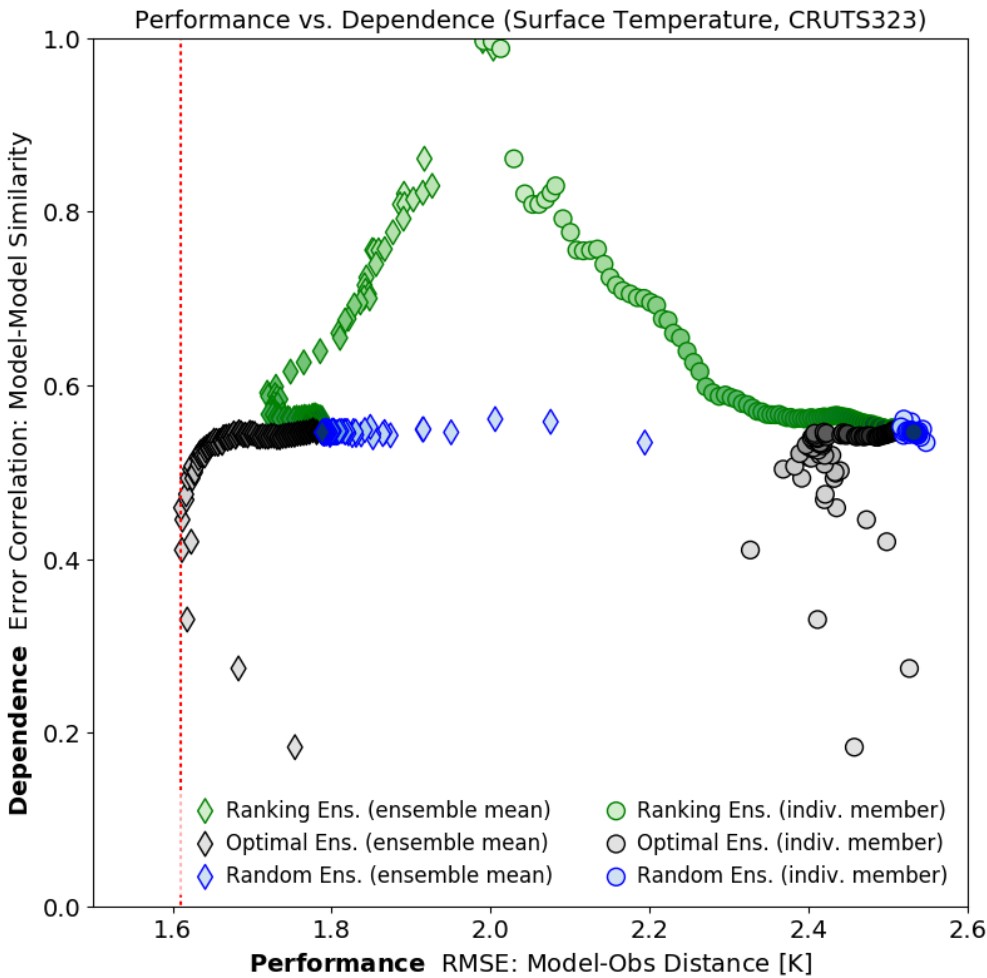

**Figure 2.** The dependence (in terms of average pairwise error correlation across all possible model pairs in the ensemble) is plotted against the performance (in terms of RMSE) for three different sampling techniques. It is based on surface air temperature and CRUTS3.23 is used as observational product. For the circular markers, the mean of model-observation distances within the ensemble is plotted against the mean of pairwise error correlations for the individual members within an ensemble for a certain ensemble size. The diamonds are used to show the RMSE of the ensemble mean (rather than the mean RMSE of the individual members) compared to the observational product. The values on the vertical axis are the same as for the circular markers. The larger the ensemble size, the darker the fill-color. The red dotted line indicates the lowest RMSE for the optimal ensemble (based on the ensemble mean).

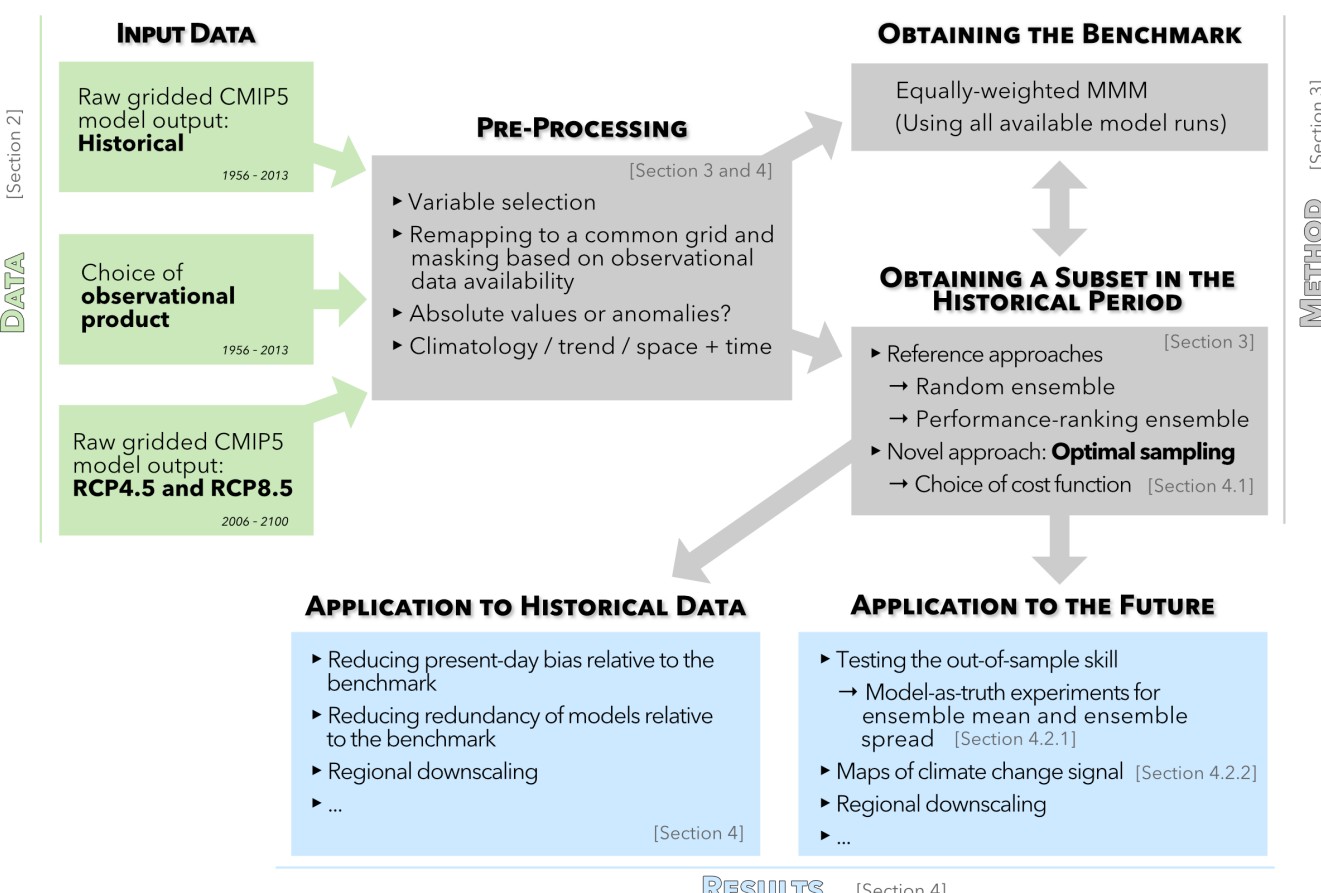

**Figure 3.** Graphical representation of the method for this study and its flexibility. The different colors are used for three sections in this publication: Data, method and results.

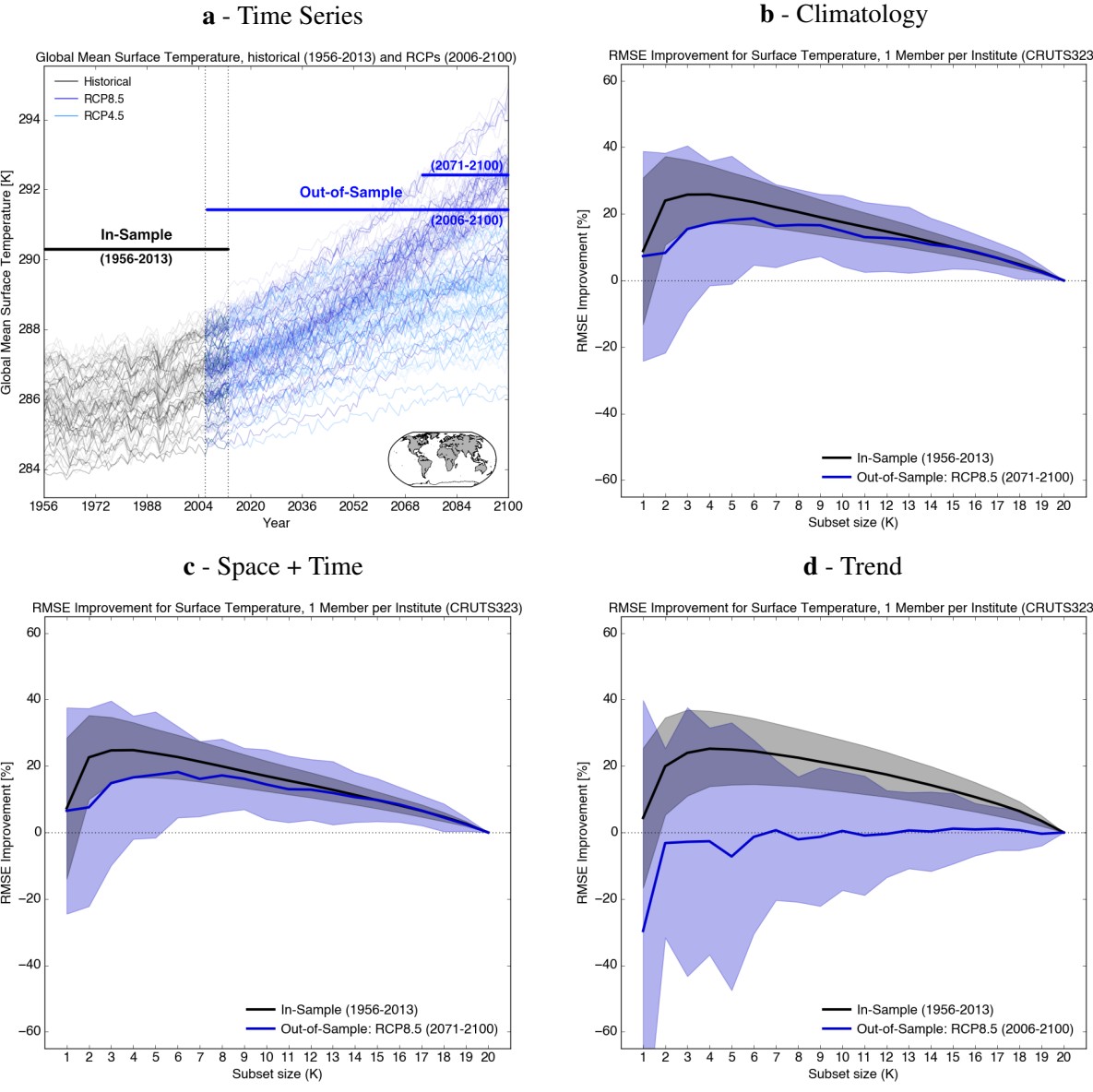

**Figure 4.** Results of the model-as-truth experiment based on three different metrics (**b-d**) and 21 model simulations (1 simulation per institute). **a**: Time series of surface air temperature averaged over the areas where CRUTS3.23 has data-availability (see map in lower right corner). The time series of the 21 model simulations which are used for the experiment are plotted slightly thicker. 1956–2013 was used as in-sample period, in which the optimal subset is derived and 2006–2100 was used as out-of-sample period for the trend metric and 2071–2100 for the remaining two metrics.

**b**: The RMSE improvement of the optimal subset relative to the MMM is plotted as a function of the subset size for each model simulation as truth. The subset for each given ensemble size was derived in the in-sample period based on the climatological metric. The curve is the mean improvement across all the 21 model simulations as "truth" and the shading around it represents the spread. Black was used for the historical period and dark blue for RCP8.5. **c** and **d** show the same as **b** but for different metrics.

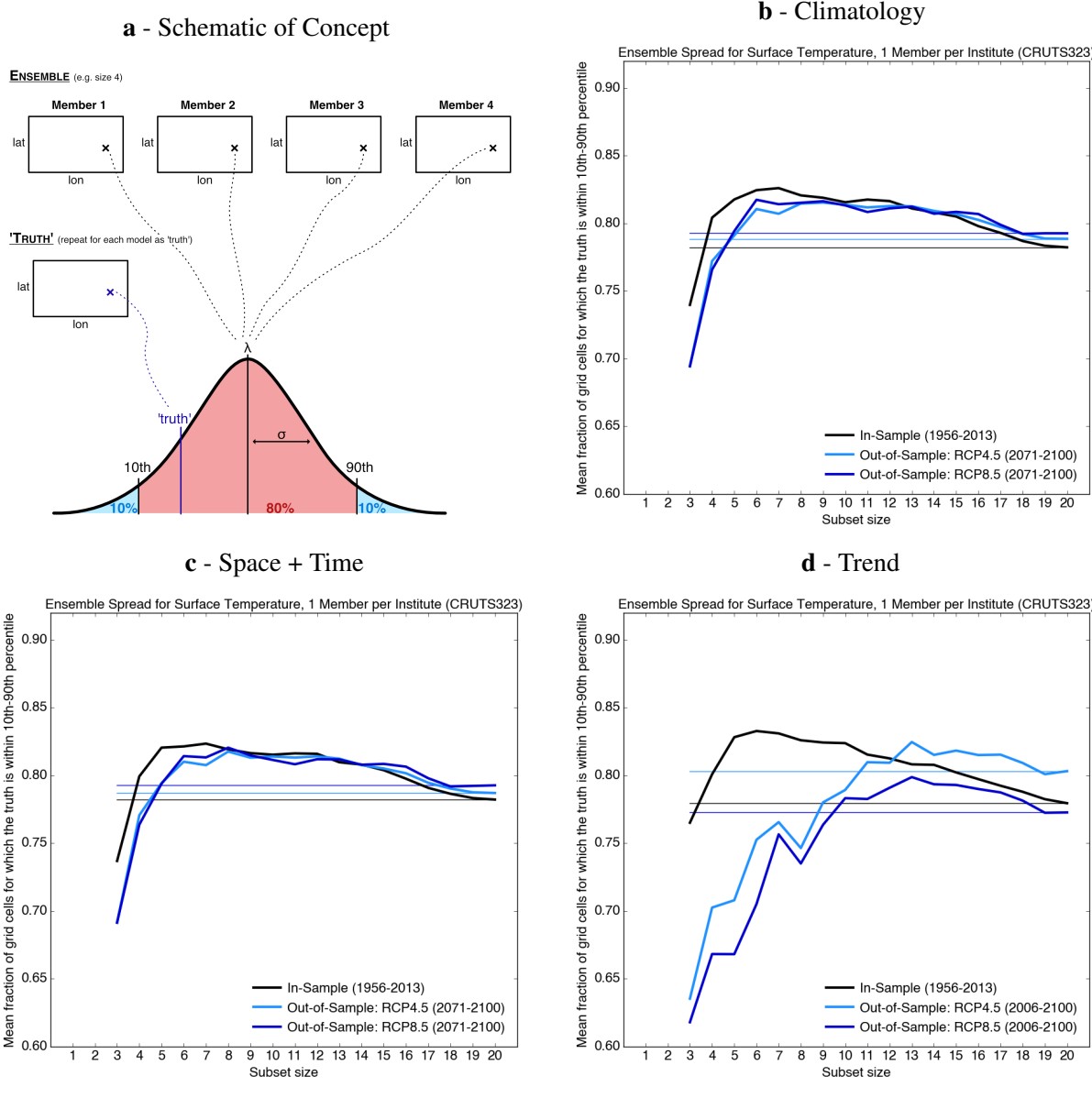

**Figure 5.** The number of times the 'model-as-truth' is within the 10th-90th percentile of ensemble spread (defined by the optimal subset for a given size) averaged across all 'truths' is plotted against the subset size. **a**: Schematic explaining how the fraction of 'truth' lying in the predicted range is obtained. **b-d**: In- (black) and out-of-sample (blue) curves for three different metrics. Surface air temperature is used as the variable. The horizontal lines refer to the percentage obtained by using all 21 model simulations.

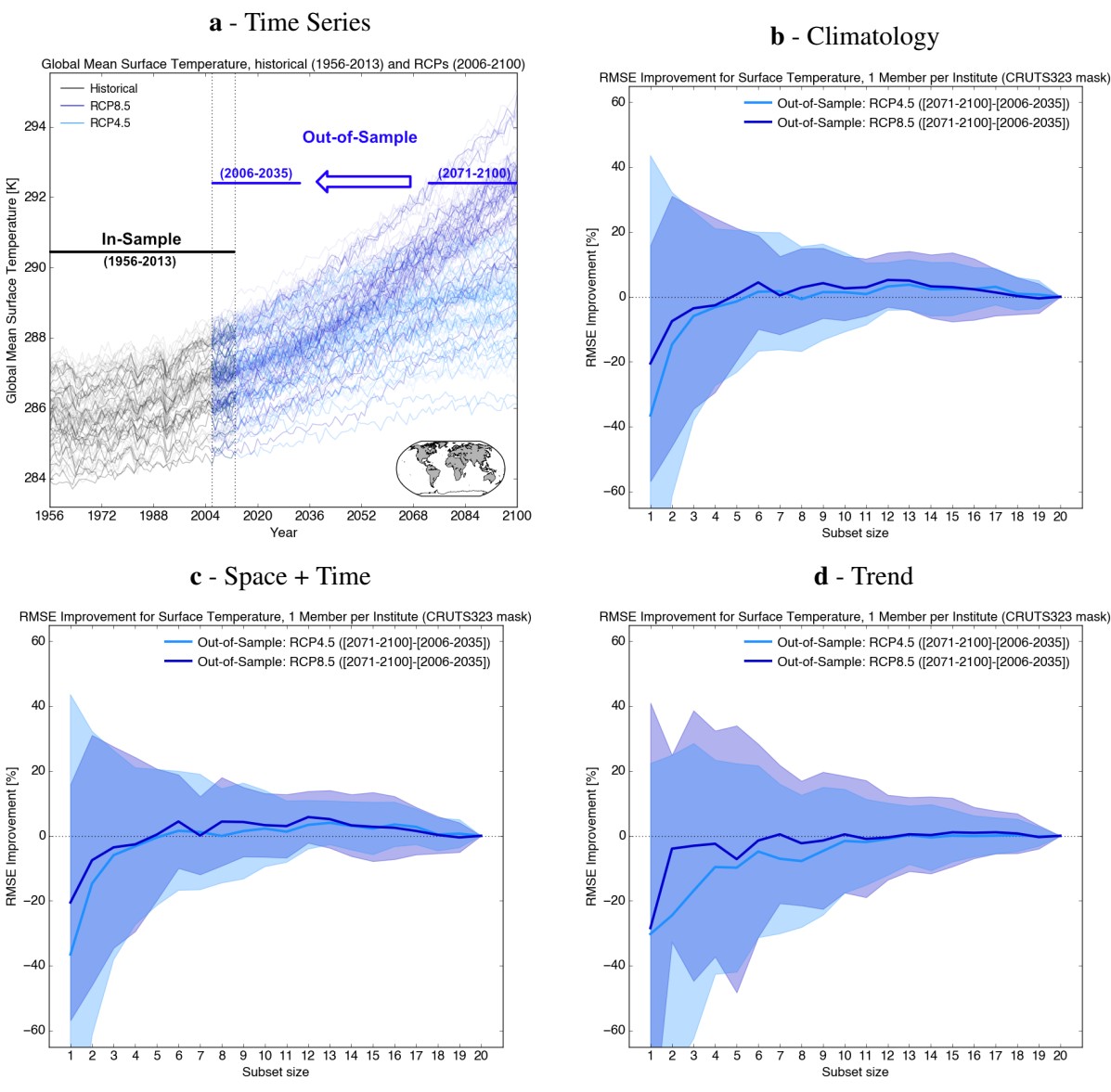

**Figure 6.** Similar to Figure 4, but here we are trying to predict the [2071–2100]-[2006–2035] temperature change (**a**) based on the optimal subsets obtained with different metrics. For **b**-**d** the optimal ensembles obtained in-sample (1956–2013) are used to predict the surface air temperature change and compared to the "true" temperature change. The same is done with the MMM and then the RMSE improvement of the optimal subset relative to the one of the MMM is calculated for both RCP4.5 and RCP8.5. The curve is the mean across all models as truth and the shading is the spread around it.

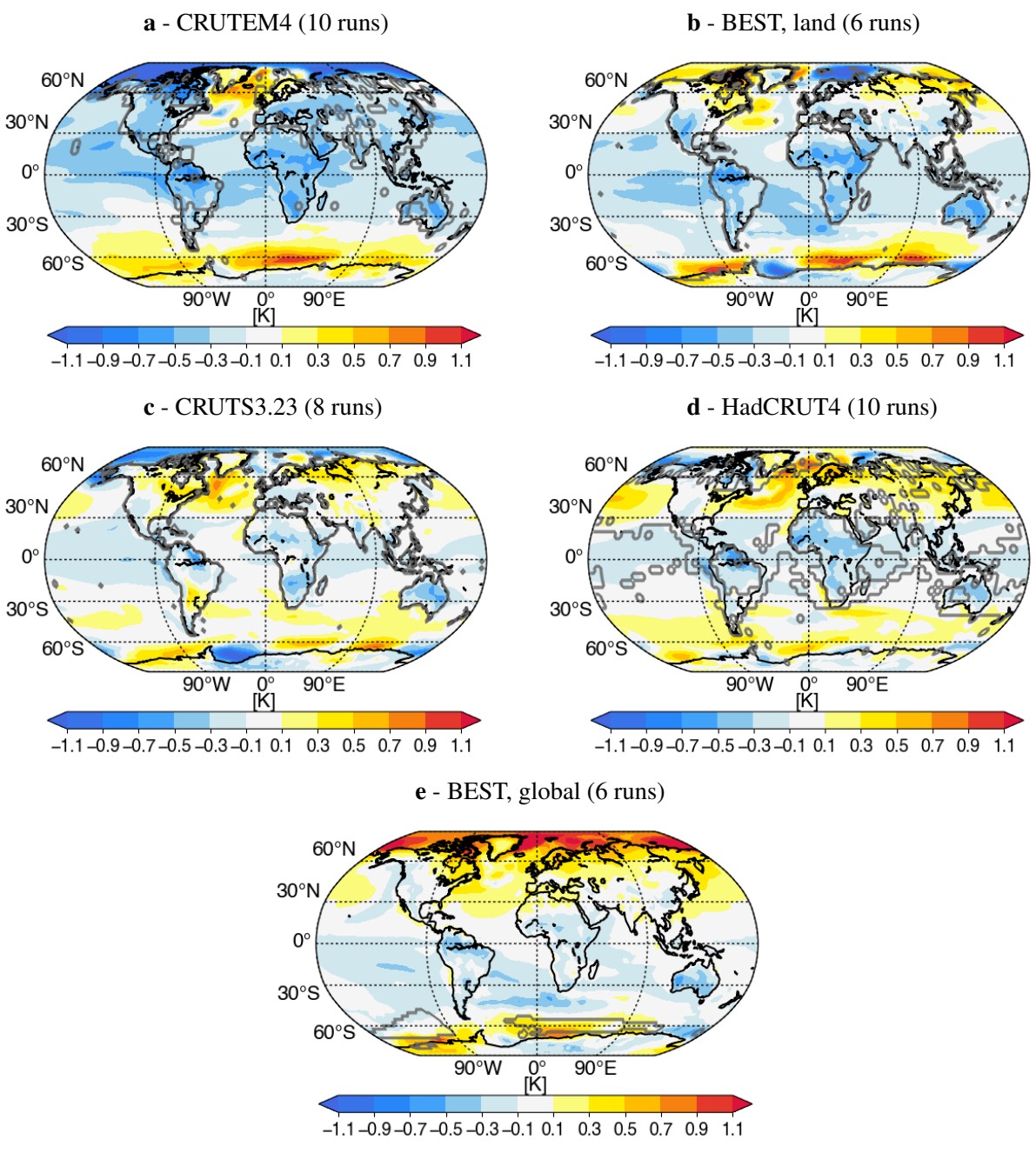

**Figure 7.** The difference between the multi-model mean (81 runs; first average across initial condition members and then averaged across 38 models) and the optimal subset is shown for the RCP8.5 surface air temperature change between [2081–2100] and [1986–2005]. The optimal subset is different depending on which observational product is used. The grey contours outline the region which was used to obtain the optimal subset in the historical period. The optimal ensemble size for each observational product is given in the title of each map.

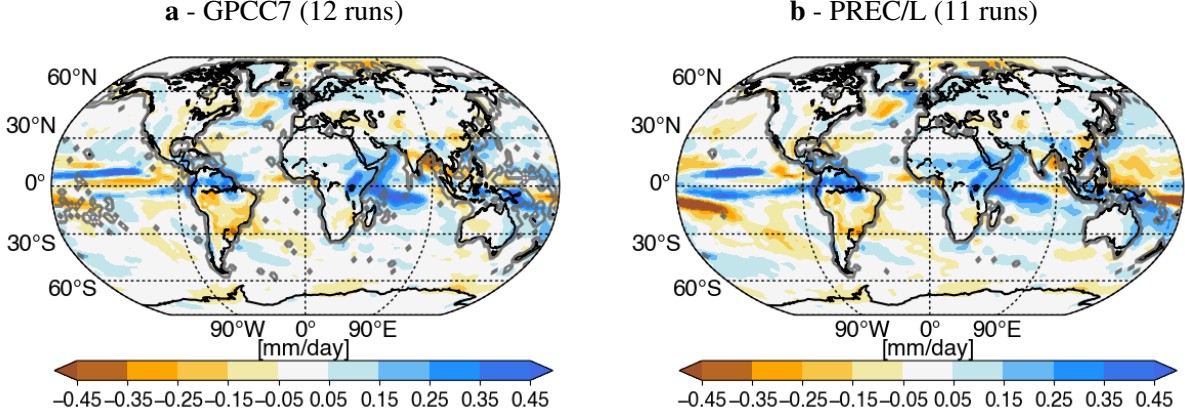

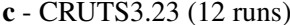

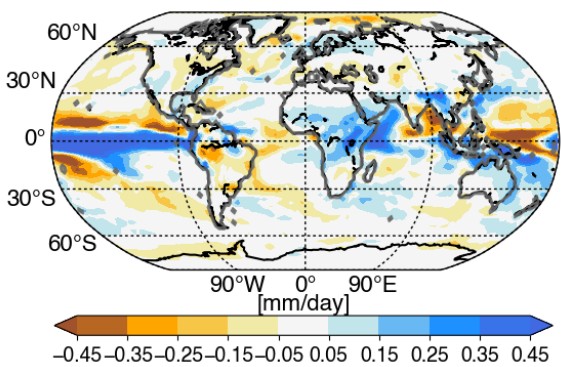

**Figure 8.** Same as Figure 7, but for precipitation change.

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
