# Peer review of "Selecting a climate model subset to optimise key ensemble properties"

_Earth System Dynamics, 2017_

## Referee Comment (RC1) · Anonymous Referee #1 · 16 May 2017

This is an interesting investigation into methods of selecting subsets of the ensemble. I think it's a useful contribution that reaches a number of conclusions that were not a priori obvious.

One weakness (which is shared with many papers) is the limited discussion of principles underlying the selection of the sub-ensemble. Having a good ensemble mean is one possible property that we might like an ensemble to have, but it's not clear whether/when/why it is important. To illustrate, if we consider a simple one-dimensional case where truth is known to take the value 5, is it better to use an ensemble of two models which take the values 3.5 and 4, or a pair with the values 0 and 9, or yet another pair with the values -10 and +20? The ensemble mean improves (relative to truth) across these three sets, but the models themselves are getting worse, which may be a concern. Another distinction between these ensembles is that both members of the

first pair share a bias in sign whereas the other two ensembles bound reality which is close to (at) the 50th percentile. I don't think these questions are easily answered but they do seem fundamental to the whole concept of how and why we use ensembles, so I think they ought to be discussed a bit more fully in the manuscript. Do the authors actually have a good argument why they would like to find an ensemble with a good mean? The analysis does also consider the issue of ensemble spread (both in model selection and assessment of the predictions) to some extent but this isn't really placed in any coherent mathematical framework. For example, the extended cost function on page 9 provides one route to distinguishing more clearly between the three different types of sub-ensembles in my example, but there does not seem to be any structured reasoning behind any particular choice.

The method of ordering by model performance seems to have some superficial similarities with Bayesian Model Averaging principles, albeit with 0-1 rather than continuous weights (and implicitly a uniform prior even when initial condition ensembles are present). It might be worth mentioning the link though I don't suppose the conclusions drawn here will be directly applicable to BMA due to the methodological differences. In particular the implied uniform prior even when IC ensembles are present would probably be considered inappropriate for any more formal implementation of BMA. On the other hand this similarity does highlight the major issue with the method, which is why the RMSE of the ensemble mean is considered to be an appropriate optimisation target in the first place. For BMA (which, at least in many artificial idealised cases, is basically the correct solution to ensemble calibration and weighting) the ensemble mean is not optimised in any meaningful sense even though it will tend to be moved towards observations in the posterior.

Despite the limitations noted above, there is clearly value in investigating the performance of different methods of ensemble selection, so I don't have any hesitation in recommending publication.

"binary": this word appears 3 times, it might be worth explaining this more fully at

the outset as meaning weights of 0 or 1 (and why this restricted choice is significant/beneficial). Actually, I believe the issue is not so much the contrast of binary (or even discrete) weights with continuous, but rather more precisely the number of zero weights, since this is what allows some models to be discarded, thereby reducing computational effort. See for example the lasso approach to regression which might have been a plausible alternative to the 0/1 methods used here. However I'm not suggesting that the authors need to investigate this as part of this piece of work.

Fig 1: The red triangles are not explained in the caption, though presumably they represent the optima from the black triangle cases.

---

## Referee Comment (RC2) · Anonymous Referee #2 · 28 Jun 2017

**General comments**

The paper presents a new method to choose an 'optimal' ensemble from a multi-model ensemble based on model performance and interdependence. The paper certainly contains many interesting aspects and presents in principle, as the authors say, important work as little has been published on this subject so far to improve policy and user relevant information from an ensemble of opportunity like CMIP.

The paper is generally well written and fits within the scope of ESD. However, the authors present this method as something that is simple to calculate and generally applicable which is by no means the case. In fact, the authors lack to clearly highlight

the aspects of their work that go beyond what has already been published. The example given as an application of their method does not seem well suited as a proof of concept to select an optimal ensemble for climate applications as it is too simple. A demonstration of how their method can be applied to multi-variable problems using multiple metrics as it would typically be needed for climate analyses would be more helpful. Another important point that is not discussed sufficiently is how to account for observational uncertainties, which is of key importance when ranking and benchmarking models. Also, even though the term 'model interdependence' is repeatedly used, no attempt is made to define model interdependence or discuss the relevant aspects for determining an optimal ensemble. Further work is required to clarify what we can learn from this study and in which cases this method can be applied, before I can recommend publication in ESD, see details below.

**Specific Comments**

I have the following major concerns which I am hoping the authors can address:

1. What is the aim of this study? Is the aim to
   (a) present a new method: then please what is new, what are the differences and advantages compared to the other methods that have recently been published (e.g., [Knutti et al., 2017; Sanderson et al., 2015a; b]? Quantitative comparisons would be required.
   (b) to present a method that is only slightly different but to provide a demonstration that this method can be used for impact studies and other climate applications? The paper fails to convincingly show that this method can be applied for concrete applications, see further comments below. The example given in the manuscript is too simple to provide any helpful insights beyond of what has already been published (see references above).

Currently a mixture of both is presented.

2. The paper could expand on recommendations of pre-selection in an ensemble. The statement on p6, l.34 that similar improvements can be made if closely related model runs are a priori removed from the ensemble to start off with a more independent ensemble could be such a recommendation.

3. It is quite confusing that within a short time this is the forth (?) recommendation for a method that should be applied for model weighting considering both model performance and interdependence (with two of the authors of this paper being also authors on all the previous papers). Yet the authors do not show the differences between this newly presented method and the previous ones. Neither they give a recommendation whether this method now supersedes the previous ones nor do they provide a sophisticated comparison of the published methods for a concrete example. For example, how would the results on sea ice extent weighting from Knutti et al. [2017] change if this method instead of the Knutti et al. [2017] method was applied and what are the policy and stakeholder relevant implications when analyzing model ensembles?

4. Related to the above: if the authors can't convincingly show what is different to the above methods, then it is also not clear what is new.

5. Climate change is not a single, but a multi-variable problem. Using RMSE as only metric does not always seem appropriate, more comprehensive metrics are available (see for example Xu et al. [2016]). The authors show that the optimal ensemble is performing best if the bias of the model subset average should be minimized - essentially indicating that the solver is working as anticipated (p6, l24). However, if a bias correction with climatological mean temperature would be the answer for an optimal ensemble, one could for example tune the models accordingly. There are good reasons why one might not want to do so (see for example Mauritsen et al. [2012]). Why would an ensemble that captures mean

temperature be better than another one? The multi-variable issue is mentioned on p7,l29 but it would be good if the authors could expand their analysis to explore this further and if possible give advice to the reader.

6. The physical consistency is mentioned yet the authors are not evaluating the optimal ensemble whether it captures other important climate features including modes of variability. This strongly limits the applications of this method and generalizations of the application like the one on p4,l10 ('We argue optimally selecting ensemble members for a set of criteria of known importance to a given problem is likely to lead to more robust projections') should be avoided.

7. Related to the above: what about model tuning? A model could be tuned towards a correct present-day temperature climatology but it might still not be the best model to project climate? What about climate sensitivity?

8. Can process-oriented diagnostics be used? This might be an interesting option to avoid selecting models that get the right results for the wrong reasons.

9. The study is motivated by the need of the impact and user community who need concrete guidance on how to use the large zoo of model output available in the CMIP ensemble (e.g. first sentence in abstract). While this is true, the paper needs to improve on giving concrete guidance. It either needs to provide real-world examples or avoid generalizations of the applicability of the method. It mathematically works fine, but whether or not it should be applied depends on whether the diagnostics chosen for the benchmark are actually relevant for the specific application. Finding these diagnostics remains a challenge.

10. The authors show that different observational products lead to different ensembles (Figure 1 and S1). But given there is observational uncertainty, some choices would need to be made. It would be good if the authors could expand

on this topic and give a recommendation how observational uncertainty can be considered in the method, the formulas presented in section 4.1 and the code.

11. Section 4.2 applies the method to the future, keeping the limited sample of weighting the ensemble based on temperature means / trends. A model could simulate a correct present-day climatology but why would it be a good model to project future climate? One of the authors convincingly shows that there is hardly any correlation between present-day and future temperature patterns [Knutti et al., 2010]. Climate change is non-linear. Could the authors choose a multi-variate and preferably process-oriented diagnostic approach? Otherwise, please limit general statements for the applicability of this method to improve projections (see above).

**Minor Comments**

There seems to be a mistake how papers are cited as they are missing 'et al.'

**References**

Knutti, R., R. Furrer, C. Tebaldi, J. Cermak, and G. A. Meehl (2010), Challenges in Combining Projections from Multiple Climate Models, J Climate, 23(10), 2739-2758, doi:doi:10.1175/2009JCLI3361.1.

Knutti, R., J. Sedláček, B. M. Sanderson, R. Lorenz, E. Fischer, and V. Eyring (2017), A climate model projection weighting scheme accounting for performance and interdependence, Geophys Res Lett, n/a-n/a, doi:10.1002/2016GL072012.

Mauritsen, T., et al. (2012), Tuning the climate of a global model, Journal of Advances

in Modeling Earth Systems, 4, doi:Artn M00a01, Doi 10.1029/2012ms000154.

Sanderson, B. M., R. Knutti, and P. Caldwell (2015a), Addressing Interdependency in a Multimodel Ensemble by Interpolation of Model Properties, J Climate, 28(13), 5150-5170, doi:10.1175/Jcli-D-14-00361.1.

Sanderson, B. M., R. Knutti, and P. Caldwell (2015b), A Representative Democracy to Reduce Interdependency in a Multimodel Ensemble, J Climate, 28(13), 5171-5194, doi:10.1175/Jcli-D-14-00362.1.

Xu, Z., Z. Hou, Y. Han, and W. Guo (2016), A diagram for evaluating multiple aspects of model performance in simulating vector fields, Geosci. Model Dev., 9(12), 4365-4380, doi:10.5194/gmd-9-4365-2016.
* * *

---

## Author Comment (AC1) · 25 Jul 2017

**Selecting A Climate Model Subset To Optimise Key Ensemble Properties - Herger et al. (2017)**

**Response to Referee #1**

We thank referee #1 for his/her valuable feedback. This document outlines our point-by-point responses to the comments made by referee #1 and the improvements we are going to make to the manuscript (*italicised text in quotation marks*).

One weakness (which is shared with many papers) is the limited discussion of principles underlying the selection of the sub-ensemble. Having a good ensemble mean is one possible property that we might like an ensemble to have, but it's not clear whether/when/why it is important. To illustrate, if we consider a simple one-dimensional case where truth is known to take the value 5, is it better to use an ensemble of two models which take the values 3.5 and 4, or a pair with the values 0 and 9, or yet another pair with the values -10 and +20? The ensemble mean improves (relative to truth) across these three sets, but the models themselves are getting worse, which may be a concern. Another distinction between these ensembles is that both members of the first pair share a bias in sign whereas the other two ensembles bound reality which is close to (at) the 50th percentile. I don't think these questions are easily answered but they do seem fundamental to the whole concept of how and why we use ensembles, so I think they ought to be discussed a bit more fully in the manuscript. Do the authors actually have a good argument why they would like to find an ensemble with a good mean? The analysis does also consider the issue of ensemble spread (both in model selection and assessment of the predictions) to some extent but this isn't really placed in any coherent mathematical framework. For example, the extended cost function on page 9 provides one route to distinguishing more clearly between the three different types of sub-ensembles in my example, but there does not seem to be any structured reasoning behind any particular choice.

We agree with the reviewer that there will never be a method that can deal with ensemble selection for all possible applications. Depending on the application, the ensemble we desire will have different properties. In some cases, finding an ensemble whose mean is close to observations might be the highest priority. This might be the case for downscaling approaches, where we want a discrete subset of models which are centered on the "truth". For impacts for example, maximising the spread in the ensemble is also desirable, as we are interested in the full spread of climate outcomes (in this case, -10 and +20 might be what we want, with models being on either side of observations). The purpose of the extended cost function is to cater for the issue of poor performing models being part of the optimal subset (see next paragraph). We added a sentence to the manuscript (Section 4.1) to address the problem of the optimal subset including poor performing models if we solely focus on optimising the RMSE between the ensemble mean and the observations: "*Also, solely focusing on the ensemble mean could potentially lead to poorer performing individual models as part of the optimal subset despite getting the mean closer to observations.*"

In other cases, we might only want the best-performing models to be part of the ensemble (here we would choose 3.5 and 4). For fields that depend on distribution shapes being representative of observations (event attribution; projections of extremes), the cost

function to minimise could be the Kolmogorov–Smirnov test statistic (this is something we are currently applying this method to).

We have identified the need for ensemble selection to be case dependent. This is why we introduce the flexibility of our cost function in the manuscript. Terms can be added to the cost function to account for different desired ensemble properties. For example, we added an additional term (Term 2 in Section 4.1, "Sensitivity to the underlying cost function") to make sure that the optimal subset does not include poor performing model runs (e.g., models of Venus or Mars will be eliminated). It is possible to add additional terms to Equation (2) to e.g., ensure that the ensemble spread is maximised, if that is an important feature of the desired subset.

We are not trying to advertise the idea of solely focussing on the mean when selecting an ensemble. This point has been discussed amongst the authors of this paper at length and we have thus tried to highlight it more prominently in the revised manuscript. We decided to show the results based on optimizing the ensemble mean because it is a conceptually simple way to illustrate this new approach. Adjusting the cost function (as done on page 9) demonstrates the flexibility of this approach for ensemble selection. For clarity, we have added the following paragraph to the section with the extended cost function:

In Section 4.1 ("Sensitivity to the underlying cost function"):
*"Reasons to use ensembles of climate models are manifold, which goes hand in hand with the need for an ensemble selection approach with an adjustable cost function. Note, that we do not consider the MSE of the ensemble mean as the only appropriate optimisation target for all applications. Even though it has been shown that the multi-model average of present day climate is closer to the observations than any of the individual model runs (e.g., Gleckler et al. (2008); Reichler and Kim (2008); Pierce et al. (2009)), it has also been shown that its variance is significantly reduced relative to observations (e.g., Knutti et al. (2010)). Also, solely focusing on the ensemble mean could potentially lead to poorer performing individual models as part of the optimal subset despite getting the mean closer to observations. Errors are expected to cancel out in the multi-model average if they are random or not correlated across models. Finding a subset whose mean cancels out those errors most effectively is therefore a good proxy for finding an independent subset, at least with respect to this metric, and is sufficient as a proof of concept for this novel approach."*

In the Introduction:
*"The aim of this study is to present a novel and flexible approach that selects an optimal subset from a larger ensemble archive in a computationally feasible way. Flexibility is introduced by an adjustable cost function which is allowing this approach to be applied to a wide range of problems."*

Section 3:
*"We then examine the sensitivity of results to observational product, cost function (to demonstrate flexibility by optimising more than just the ensemble mean) and other experimental choices."*

The method of ordering by model performance seems to have some superficial similarities with Bayesian Model Averaging principles, albeit with 0-1 rather than continuous weights (and implicitly a uniform prior even when initial condition ensembles are present). It might be worth mentioning the link though I don't suppose the conclusions drawn here will be directly applicable to BMA due to the methodological differences. In particular the implied

uniform prior even when IC ensembles are present would probably be considered inappropriate for any more formal implementation of BMA. On the other hand this similarity does highlight the major issue with the method, which is why the RMSE of the ensemble mean is considered to be an appropriate optimisation target in the first place. For BMA (which, at least in many artificial idealised cases, is basically the correct solution to ensemble calibration and weighting) the ensemble mean is not optimised in any meaningful sense even though it will tend to be moved towards observations in the posterior.

The idea of BMA is certainly similar as it also tries to solve the problem of model selection and combined estimation. However, the difference between 0/1 and continuous weights is central in this case. Also, the model weights in BMA would be derived from performance (model's capability to accurately describe the data) only. As we have shown in our study, solely accounting for performance in ensemble selection is not recommended and can even be worse than a random ensemble. This probably has implications for the BMA approach.

As noted above, optimising for the ensemble mean as presented in the manuscript was a conceptually simple approach to illustrate the technique and we do not claim that solely focussing on the mean is desirable for all cases. Hence the addition of the extended cost function. Hopefully our additional text has clarified this a little.

"binary": this word appears 3 times, it might be worth explaining this more fully at the outset as meaning weights of 0 or 1 (and why this restricted choice is significant/beneficial). Actually, I believe the issue is not so much the contrast of binary (or even discrete) weights with continuous, but rather more precisely the number of zero weights, since this is what allows some models to be discarded, thereby reducing computational effort. See for example the lasso approach to regression which might have been a plausible alternative to the 0/1 methods used here. However I'm not suggesting that the authors need to investigate this as part of this piece of work.

We agree with the reviewer that we should be clearer on the meaning of the word "binary" in this context. We have added a sentence to the main text, also highlighting that we contrast binary with continuous weights mostly because of the zero weight. For clarity, we have added the following two sentences to the Introduction:
*"With binary we refer to the weights being either zero or one, and thus a model run is either discarded or part of the subset."*

*"More precisely, it is the number of zero weights that leads to some models being discarded from the ensemble."*

The Lasso approach is certainly an interesting option due to assigning weights of 0 to some model runs, however it is, to our understanding, not possible to adjust the cost function that is being minimised (by default: RMSE). The optimizer we are using in our work allows us to define constraints and cost functions depending on the use-case. We regard the flexibility of being able to adjust the cost function depending on the aim of the study as an important strength of our method. We added the following text (Section 5):

*"The lasso regression analysis method (Tibshirani, 2011) often used in the field of machine learning tries to select a subset of features (in our case: model simulations) to improve prediction accuracy. It is similar to the presented approach in a way that it also*

*selects a subset of models by applying weights of zero. However, contrary to the method presented here, it is to our knowledge not possible to customise the cost function that is being minimised (by default: RMSE)."*

Fig 1: The red triangles are not explained in the caption, though presumably they represent the optima from the black triangle cases.

The reviewer is correct. We have now added the following explanation of the red triangles to the caption of Figure 1:
*"The corresponding red triangle is the optimal subset of the black triangle cases."*

---

## Author Comment (AC2) · 25 Jul 2017

**Selecting A Climate Model Subset To Optimise Key Ensemble Properties - Herger et al. (2017)**

**Response to Referee #2**

We thank referee #2 for taking the time to review our manuscript. This document outlines our point-by-point responses to the comments made by referee #2 and the improvements we are going to make to the manuscript (*italicised text in quotation marks*).

The paper is generally well written and fits within the scope of ESD. However, the authors present this method as something that is simple to calculate and generally applicable which is by no means the case. In fact, the authors lack to clearly highlight the aspects of their work that go beyond what has already been published. The example given as an application of their method does not seem well suited as a proof of concept to select an optimal ensemble for climate applications as it is too simple. A demonstration of how their method can be applied to multi-variable problems using multiple metrics as it would typically be needed for climate analyses would be more helpful. Another important point that is not discussed sufficiently is how to account for observational uncertainties, which is of key importance when ranking and benchmarking models. Also, even though the term 'model interdependence' is repeatedly used, no attempt is made to define model interdependence or discuss the relevant aspects for determining an optimal ensemble. Further work is required to clarify what we can learn from this study and in which cases this method can be applied, before I can recommend publication in ESD, see details below.

We thank the reviewer for his/her comments. We note the lack of clarity in the Introduction, which when addressed should answer a few of the reviewer's concerns (see below and other responses to reviewer concerns in this document). It is important to highlight that there is no single best approach for ensemble selection available and our approach does not replace any of the other techniques in the literature. Any approach will have to be tailored depending on the specific use-case. Using Gurobi offers the ability to customise the cost function and metrics used for obtaining an optimal subset. This is essential for a given approach to be widely applied. Attempting to find a single best approach is therefore a pointless task; hence our focus on finding an approach that could potentially be applied to a wide range of use-cases.

We explain the range of approaches for model weighting that have recently emerged with the range of applications that such an approach can be applied to. Bishop & Abramowitz (2013) for example focus their approach solely on variance by looking at time series and finding a linear combination of model runs to most accurately represent observational variability. Sanderson et al. (2015) however focus on climatology without considering any time component. Just as there are many ways of addressing model performance, there are many ways of addressing independence.

The text in the Introduction was adjusted to make this clearer:
*"[...] The same process was also used for future projections, with the danger of overfitting mitigated through out-of-sample performance in model-as-truth experiments (Abramowitz and Bishop, 2015). In their approach, they solely focus on variance by looking at time*

*series. Another method also using continuous weights but considering climatologies rather than time series was proposed by Sanderson et al. (2015a). It is based on dimension reduction of the spatial variability of a range of climatologies of different variables. [...]"*

The reviewer rightly comments that we did not define model interdependence. This is because the definition of dependence is problem-dependent. Most of the authors on this manuscript attended a workshop last December on exactly this topic where it became evident that a generally agreed-on definition is currently absent.

Rather than testing our approach on multiple variables at a time we did it separately for surface air temperature and total precipitation. Monthly mean temperature is a variable commonly used by the community, and the problem at hand (e.g. one model one vote) has been clearly framed in other work by some authors on this paper.

We discuss the topic of observational uncertainty below in our answer to Q10.

**1.** What is the aim of this study? Is the aim to (a) present a new method: then please what is new, what are the differences and advantages compared to the other methods that have recently been published (e.g., [Knutti et al., 2017; Sanderson et al., 2015a; b]? Quantitative comparisons would be required. (b) to present a method that is only slightly different but to provide a demonstration that this method can be used for impact studies and other climate applications? The paper fails to convincingly show that this method can be applied for concrete applications, see further comments below. The example given in the manuscript is too simple to provide any helpful insights beyond of what has already been published (see references above).
Currently a mixture of both is presented.

We note the lack of clarity in our framing of the contribution this work makes, and have therefore adjusted the Introduction accordingly:

*"The aim of this study is to present a novel and flexible approach that selects an optimal subset from a larger ensemble archive in a computationally feasible way. Flexibility is introduced by an adjustable cost function which is allowing this approach to be applied to a wide range of problems."*

*"Such an approach with binary (0/1) rather than continuous weights is desired to obtain a smaller subset that can drive regional models for impact studies, as this is otherwise a computationally expensive task."*

The aim of this manuscript is mainly the reviewer's (a). We are presenting a new, flexible ensemble selection method that can be applied to impact studies. It is not clear to us that in order to address point (a), a quantitative comparison to previous approaches is required. Comparing existing approaches for a given use-case is certainly something valuable that should be done in the future, but it goes beyond the scope of this study, given that detailing the technique alone has already made this manuscript reasonably long.

We also think that introducing a new approach, as stated in (a) without showing where it could be applied would not be very useful. We therefore also touch on (b) by highlighting that such an approach could be used for impact studies which requires a small number of runs (e.g. for dynamical downscaling). This point has been addressed in the introductory

part of the manuscript, see here:

*"Regional dynamical downscaling presents a slightly different problem to the one stated above, as the goal is to find a small subset that reproduces certain statistical characteristics of the full ensemble. In this case the issue of dependence is critical, and binary weights are needed, since computational resources are limited."*

As our approach results in a discrete subset, we do not see the need to perform the additional step of using this optimal subset for downscaling and impact assessment. The novelty is to find a discrete optimal subset for a given use-case, and thus using that for impact studies would add little to the literature and goes beyond the scope of this study.

We believe the Introduction already covers the main differences between this approach and existing approaches.

Theoretically, a (c) could be added to our aim: making it clear that asking 'which of the existing approaches is the best' is not a well framed question. It is equivalent to asking 'which climate model is the best?', without specifying the application. Only when calibrated to a given use-case it is useful to compare existing approaches of ensemble selection, or definitions of model dependence.

**2.** The paper could expand on recommendations of pre-selection in an ensemble. The statement on p6, l.34 that similar improvements can be made if closely related model runs are a priori removed from the ensemble to start off with a more independent ensemble could be such a recommendation.

One conclusion that emerged from the workshop on model dependence in multi-model climate ensembles, held in December 2016, was the idea to write a review paper on this topic. The participants are currently working on a review of the current literature around this topic and are trying to give recommendations on how to use multi-model ensembles whose members are not independent.

Pre-selection in the ensemble will always be somewhat subjective and case-dependent. Giving general recommendations of pre-selection in an ensemble is thus not straightforward. We have, however, added the following sentence regarding the possibility of filtering out certain model runs before starting the optimization process (Section 4.1, "Sensitivity to the underlying cost function"):

*"It would of course also be possible to make an a priori decision on which models should be considered before starting the optimisation process."*

**3.** It is quite confusing that within a short time this is the forth (?) recommendation for a method that should be applied for model weighting considering both model performance and interdependence (with two of the authors of this paper being also authors on all the previous papers). Yet the authors do not show the differences between this newly presented method and the previous ones. Neither they give a recommendation whether this method now supersedes the previous ones nor do they provide a sophisticated comparison of the published methods for a concrete example. For example, how would the results on sea ice extent weighting from Knutti et al. [2017] change if this method instead of the Knutti et al. [2017] method was applied and what are the policy and stakeholder relevant implications when analyzing model ensembles?

We hope that our answer to the reviewer's first comment already addresses some of those

concerns. As mentioned before, there is no single best approach. Which approach to choose depends on the the specific use case. In some cases (e.g., when simply computing a mean and range across a set of GCMs), continuous weights are sufficient. In others, having a discrete subset of models is appropriate, e.g., for subsequent downscaling, because dynamical downscaling is computationally expensive and can thus only be applied to a small subset of model runs.

The reviewer mentioned the Arctic sea ice extent weighting from Knutti et al. (2017). This is an example where the benefit of model weighting (compared to simply taking the equally-weighted multi-model mean) is expected to be very large as some models are not even able to capture the present day state properly. Global mean temperatures are usually captured more accurately by models than sea ice extent and if we see improvement in the ensemble mean in this case, we regard this as a stronger proof of concept. We therefore do not see the need to apply our method to this exact use-case.

The introduction states the main differences between the existing approaches. However, for clarity we have added a few sentences to the Introduction to make this clearer (see also below Q4):

*"This approach is not meant to replace or supersede any of the existing approaches in the literature. Just as there is no single best climate model, there is no universally best model weighting approach. Whether an approach is useful depends on the criteria that are relevant for the application in question. Only once the various ensemble selection approaches have been tailored to a specific use-case, can a fair comparison be made. Flexibility in ensemble calibration by defining an appropriate cost function that is being minimised and metric used is key for this process."*

**4.** Related to the above: if the authors can't convincingly show what is different to the above methods, then it is also not clear what is new.

The main difference between this approach and most of the existing ones is the use of binary (zero or one) weights rather than continuous weights. Having a zero weight leads to a discrete subset which can subsequently be used for regional downscaling (and used for impact studies) — desirable as computational cost is then reduced compared to if one would use the full ensemble. Note, that the stepwise model elimination procedure described in Sanderson et al. (2015) can also be considered to be an approach with binary weights. It is different from what we did as the focus is on joint projections of multiple variables and is arguable more technically challenging to implement.

Apart from having a discrete subset, the method allows for changes in the cost function being optimised and the metric used. Different from most other approaches, out-of-sample performance has been tested to avoid overfitting of the ensemble to the present-day state. Also, by providing the code, we see no reason why it would be much of a hurdle to implement. Other published approaches are considerably more technically challenging (e.g. Sanderson et al. (2015), Bishop and Abramowitz (2013)).

To make this clearer, we added a few sentences to the Introduction of the manuscript (see above).

*"This approach is not meant to replace or supersede any of the existing approaches in the literature. Just as there is no single best climate model, there is no universally best model weighting approach. Only once the various ensemble selection approaches have been tailored to a specific use-case, can a fair comparison be made. Flexibility in ensemble*

*calibration by defining an appropriate cost function that is being minimised and metric used is key for this process."*

We have also added the following paragraph to Section 4.1 ("Sensitivity to the underlying cost function"):

*"Reasons to use ensembles of climate models are manifold, which goes hand in hand with the need for an ensemble selection approach with an adjustable cost function. Note, that we do not consider the MSE of the ensemble mean as the only appropriate optimisation target for all applications. Even though it has been shown that the multi-model average of present day climate is closer to the observations than any of the individual model runs (e.g., Gleckler et al. (2008); Reichler and Kim (2008); Pierce et al. (2009)), it has also been shown that its variance is significantly reduced relative to observations (e.g., Knutti et al. (2010)). Errors are expected to cancel out in the multi-model average if they are random or not correlated across models. Finding a subset whose mean cancels out those errors most effectively is therefore a good proxy for finding an independent subset, at least with respect to this metric, and is sufficient as a proof of concept for this novel approach."*

**5.** Climate change is not a single, but a multi-variable problem. Using RMSE as only metric does not always seem appropriate, more comprehensive metrics are available (see for example Xu et al. [2016]). The authors show that the optimal ensemble is performing best if the bias of the model subset average should be minimized - essentially indicating that the solver is working as anticipated (p6, l24). However, if a bias correction with climatological mean temperature would be the answer for an optimal ensemble, one could for example tune the models accordingly. There are good reasons why one might not want to do so (see for example Mauritsen et al. [2012]). Why would an ensemble that captures mean temperature be better than another one? The multi-variable issue is mentioned on p7,l29 but it would be good if the authors could expand their analysis to explore this further and if possible give advice to the reader.

The reviewer is correct that climate change cannot be fully addressed by solely looking at one variable or metric. However, this is not what we are trying to accomplish with this work. Note that we optimize spatial fields not global means, and the former cannot really be tuned in a GCM.
To introduce this novel approach, we separately applied it to surface temperature and total precipitation, using RMSE as a metric. It can of course be applied to more variables, as long as reliable observations are available, and once suitable scaling factors are chosen to aggregate different units. As long as it can be implemented into the solver Gurobi, almost any other metric of interest is possible. For example, we have begun working on a related project using the Kolmogorov–Smirnov test statistic instead of RMSE (reducing distribution biases which is for example relevant for event attribution). We expanded the paragraph with the following text where we talk about the multi-variable issue to make the flexibility of this approach clearer (Section 4.1, "Sensitivity to the underlying cost function"):

*"The cost function presented in this study solely uses MSE as a performance metric. There are of course many more metrics available (e.g. Xu et al. (2016), Taylor (2001), Gleckler et al. (2008), Baker and Taylor (2016)) that we might choose to implement in this system for different applications. So as not to confuse this choice with the workings of the ensemble selection approach, however, we illustrate it with RMSE alone, as this is what most existing approaches in this field use to define their performance weights (e.g. Knutti et al. (2017), Sanderson et al. (2017), Abramowitz and Bishop (2015))."*

We note that when comparing panels (a) and (b) in Figure 1, depending on the chosen variable, we end up with a different optimal ensemble size, different ensemble members and different performance gains. This is best framed as a calibration exercise since one can only obtain an optimal subset for a clearly defined use-case (given the variable, metric, region, observational product etc.).

If the goal is to obtain a single optimal subset across multiple variables, one could preprocess the model output in a way Sanderson et al. (2015) did in their Journal of Climate paper (see their Figure 1). Gridded model output is normalized and concatenated into a long multi-variable vector which is then used for further analysis where a single cost function is optimized. We added a few sentences to our manuscript highlighting the possibility of doing the same (see below). Even though this will result in a single optimal subset across all variables, it is sensitive to how the variables were normalized and it also conceals the fact that the optimal subset for the individual variables might look very different. In many cases it is therefore useful to employ the calibration exercise on each variable separately to see how the optimal subset varies instead of first combining all the variables and then finding a single optimal subset. Additionally, if only one variable is of interest for a particular case, one can only gain from selecting a subset based on only that variable. The following text has been added (Section 4.1, "Variable choice"):

*"This could most simply be done using a single cost function that consists of a sum of standardised terms for different variables. This is similar to what has been done in Sanderson et al. (2015a) (see their Figure 1). However, this might conceal that fact that the optimal subsets for the individual variables potentially look very different. "*

Alternatively, a Pareto solution set of ensembles is possible, which is often used in multicriteria calibration papers for hydrological models. For example: Gupta et al. (1998): "Toward improved calibration of hydrologic models: Multiple and noncommensurable measures of information".

**6.** The physical consistency is mentioned yet the authors are not evaluating the optimal ensemble whether it captures other important climate features including modes of variability. This strongly limits the applications of this method and generalizations of the application like the one on p4,l10 ('We argue optimally selecting ensemble members for a set of criteria of known importance to a given problem is likely to lead to more robust projections') should be avoided.

Given that we are not assigning continuous weights to the CMIP5 ensemble member, our subset is as physically consistent as the original ensemble. While a model average may not show physically plausible behaviour, each single model run should (to the degree that it represents the real world), and using each individually for impact analysis or downscaling will preserve as much of the physical consistency as possible.

It is true that the cost functions we have used to illustrate the technique are simple, not comprehensive, and in particular not focused on modes of climate variability. This work is only a first step, being the introduction of a new method. There are a myriad of modes of variability that one could attempt to calibrate an ensemble towards. Which ones are important? Again, it comes down to the specific use-case (e.g. region, variable, question.)

**7.** Related to the above: what about model tuning? A model could be tuned towards a correct present-day temperature climatology but it might still not be the best model to

project climate? What about climate sensitivity?

We agree with the reviewer that a model which is very closely tuned toward the present-day state won't necessarily be skillful for projections. The model-as-truth experiment in our paper is intended to check for the possibility of overfitting/"over-tuning" of the ensemble members to the present state, although it could perhaps be better explained. We added the following to better motivate the use of the model-as-truth experiment (Section 4.2.1):

*"Rigid model tuning for example could cause the ensemble to be heavily calibrated on the present-day state. An optimal subset derived from such an ensemble would not necessarily be skillful for future climate prediction as we are dealing with overfitting and we are not calibrating to biases that persist into the future. This is where model-as-truth experiments come into play."*

Note that while global mean temperature can be tuned to some degree, the spatial fields of climatology cannot (otherwise the current GCMs would not have such large persistent climatological biases). Regarding climate sensitivity, climate models which are biased high (in terms of temperature for example) in present day, are often at the higher end of the distribution in the projections. In our approach, we make use of this persistent bias. Improvement of our optimal subset relative to the ensemble mean (of 1 run per institute) is expected to decrease with increasing time/forcing, as the climate system will reach a state it has never experienced before. At that point, calibrating a subset on the present day might not lead to any improvement. However, this is a problem for any weighting or calibration approach, and one way to check for this is to use model-as-truth experiments to show where we have a breakdown of predictability. This is certainly something worth exploring in a future study.

However, for what we have used it for, there seems to be some predictability in the system and using the optimal subset out-of-sample is likely to have advantages over simply using the equally-weighted multi-model mean. For clarity, we have added the following text (Section 4.2.1):

*"Climate models which are biased high (in terms of temperature for example) in the present day, are often at the higher end of the distribution in the projections. This is related to climate sensitivity and our approach is able to make use of this persistent bias."*

In Section 5:
*"Using model-as-truth experiments, we observed that the skill of the optimal subset relative to the unweighted ensemble mean decreases the further out-of-sample we were testing it. This breakdown of predictability is not unexpected as the climate system reached a state it has never experienced before. This is certainly an interesting aspect which should be investigated in more depth in a future study."*

**8.** Can process-oriented diagnostics be used? This might be an interesting option to avoid selecting models that get the right results for the wrong reasons.

This is an interesting point, which also came up in discussions among the authors of this manuscript. Depending on the application, process-oriented diagnostics can potentially improve the ensemble selection by giving us more confidence of selecting the subset for the right reasons. We decided to focus on global temperature and precipitation as this

manuscript is a proof of concept, and introducing the ensemble selection approach for another specific example might be confusing. Also, multiple observational products exist for those two variables and sensitivity to the chosen product could be tested.

The metric used for this approach can take any form which makes it very flexible. A few sentences have been added to the manuscript to highlight the possibility of using process-oriented diagnostics (Section 4.1, "Variable choice"):

*"The presented approach can obtain an optimal subset for any given variable, as long as it is available across all model runs and credible observational products exist. One might even consider using process-oriented diagnostics to provide greater confidence when selecting a subset for the right physical reasons."*

**9.** The study is motivated by the need of the impact and user community who need concrete guidance on how to use the large zoo of model output available in the CMIP ensemble (e.g. first sentence in abstract). While this is true, the paper needs to improve on giving concrete guidance. It either needs to provide realworld examples or avoid generalizations of the applicability of the method. It mathematically works fine, but whether or not it should be applied depends on whether the diagnostics chosen for the benchmark are actually relevant for the specific application. Finding these diagnostics remains a challenge.

Given that our approach results in a discrete subset, using it subsequently for regional downscaling and then impact assessments is certainly an application we had in mind. However, we do not agree with the reviewer that it should be within the scope of this study to give an impacts-replated example. As the reviewer states, it "mathematically works fine" and this is what we wanted to demonstrate here (proof of concept). The novel part is finding a discrete subset of model runs and the subsequent steps needed for impact assessments are unchanged and thus do not need to be discussed here in detail.

We have adjusted the manuscript to highlight the importance of tailoring the cost function and metric to the problem at hand to avoid generalizations, as noted above.

**10.** The authors show that different observational products lead to different ensembles (Figure 1 and S1). But given there is observational uncertainty, some choices would need to be made. It would be good if the authors could expand on this topic and give a recommendation how observational uncertainty can be considered in the method, the formulas presented in section 4.1 and the code.

We agree with the reviewer that text could be added discussing the problem of observational uncertainty. However, no single best solution exists for this problem. Before starting the calibration (i.e. ensemble selection) exercise, one should first identify which observational products can be trusted (for the specific region, variable, time period in mind).

The discussion is actually similar to the reviewer's question 5 (for multiple observational products instead of variables). In this study, we presented a different optimal subset for each chosen observational product. Alternatively, one could of course put multiple observational products into a single cost function and end up with a single optimal subset. However, when using ensembles for inference, then a lot can be learned about predictability from the differences between using different observational products. This additional uncertainty added by observations is ignored if all the products are combined in a single cost function.

As for Q5, one could also end up with a pareto front across different products, where we have a whole range of subsets rather than a single best one. This is something that is worth investigating in a future study. The following text has been added (Section 4.1, "Choice of observational product"):

*"This could be done by putting multiple observational products into a single cost function. However, when using ensembles for inference, a lot can be learned from the spread across observational products. This additional uncertainty added by observations is ignored if all the products are combined in a single cost function."*

**11.** Section 4.2 applies the method to the future, keeping the limited sample of weighting the ensemble based on temperature means / trends. A model could simulate a correct present-day climatology but why would it be a good model to project future climate? One of the authors convincingly shows that there is hardly any correlation between present-day and future temperature patterns [Knutti et al., 2010]. Climate change is non-linear. Could the authors choose a multivariate and preferably process-oriented diagnostic approach? Otherwise, please limit general statements for the applicability of this method to improve projections (see above).

In order to test if the subset has skill in the future (we call it out-of-sample, as we do not have observations), we conducted model-as-truth experiments. From that we learned that our optimal subset does not always improve projections relative to the simple multi-model mean, especially when optimizing for the trend (Fig. 4d). When optimizing for the climatology however, we observe an improvement of more than 10% out-of-sample. This suggests that we are not simply fitting noise, but actually gaining from the subset selection. If there was no signal in the present-day climatology, we would not have obtained an improvement out-of-sample.
We agree with the reviewer that correlations between present-day and future temperature patterns are weak (see also our supplementary figure S5). Finding a good emergent constraint is exactly what is needed to find an optimal subset with skill out-of-sample. Regional biases seem to persist, which is why we found improved out-of-sample skill in some cases.

We commented on the idea of using process-oriented diagnostics at Q8 (above).

We added a few sentences at the beginning of Section 4.2.1 to better motivate the need for model-as-truth experiments.

*"Is a model that correctly simulates the present-day climatology automatically a good model for future climate projections? To answer this question, we need to investigate if regional biases persist into the future, and determine whether the approach is fitting short term variability. This is done by conducting model-as-truth experiments."*

Minor Comment: There seems to be a mistake how papers are cited as they are missing 'et al.'

We have adjusted the bibliography so that the "et al." are now shown in the revised version of the manuscript.

---

## Author Response (AR2)

**Selecting A Climate Model Subset To Optimise Key Ensemble Properties  -  Herger et al. (2017)**

**Report #1**

We thank the reviewer for taking the time to review our manuscript for a second time and for providing valuable feedback. This document outlines our point-by-point responses to the comments made by the reviewer and the improvements we have made to the manuscript (*italicised text in quotation marks*).

> The authors have addressed most of my comments. Overall, they made some minor changes to the manuscript in response to the reviews.
> However, they have not addressed my main comment, which is that using a present-day mean climatology of temperature to weight temperature projections seems not a good idea since there is hardly any correlation between present-day and future temperature patterns (see Knutti et al., 2010). The authors say in the abstract: "Here, we present an efficient and flexible tool that makes better use of the ensemble as a whole by finding a subset with improved mean performance compared to the multi-model mean while at the same time maintaining the spread and addressing the problem of model interdependence." I doubt that an ensemble weighted by mean temperature is necessarily a better ensemble for impact studies. While the mean can be correct the response to forcing could be very poorly represented in a model for example. I can only recommend publication when at least a time-varying diagnostic is additionally included. Alternatively this should be presented as a new method which clearly requires more work on the diagnostic side before it is actually be applied to weight models. Then sentences like the one above should be changed.
>
> References:
>
> Knutti, R., R. Furrer, C. Tebaldi, J. Cermak, and G. A. Meehl (2010), Challenges in Combining Projections from Multiple Climate Models, J Climate, 23(10), 2739-2758, doi:doi:10.1175/2009JCLI3361.1.

We are glad to hear that we managed to address most of the reviewer's comment from the first round.

We also agree with the reviewer's one remaining concern that investigating a time-varying diagnostic is a good idea. Indeed we already consider two different time-varying diagnostics in the manuscript: optimising for space and time varying fields in one instance, and trend in a separate application. Those diagnostics are introduced in Section 4.1 (in the "Alternatives to climatology" paragraph), which the reviewer must have missed. We have therefore expanded this paragraph to better introduce those diagnostics. Results for those diagnostics in addition to the climatology diagnostic are shown throughout the manuscript (Figures 4-6). The paragraph now reads as follows:

*"As part of the data pre-processing step, we computed climatologies (time-means at each grid cell) for the model output and observational dataset. In addition to climatologies, we decided to consider time-varying diagnostics ("trend" and "space+time"), which potentially contain information relevant for projections, which is not captured by time-means. For the "trend" diagnostic, we compute a linear trend of the corresponding variable at each grid cell and end up with a two dimensional array for each simulation and observational dataset. As a second time-varying diagnostic, we compute 10-year running means at each grid-cell to obtain a three dimensional array which is subsequently used for the analysis. This is hereafter referred to as "space+time". Subsection 4.2.1 shows (based on a model-as-truth experiment) how sensitive the ensemble can be to the diagnostics (mean, trend, or variability) chosen in the pre-processing step."*

Hopefully this is enough to address the reviewer's concern that "*I can only recommend publication when at least a time-varying diagnostic is additionally included.*"

The reviewer writes: "...*using a present-day mean climatology of temperature to weight temperature projections seems not a good idea...*". We cannot fully agree with the statement that there is no value in future projections that have decreased climatology bias. Based on a model-as-truth experiment, Figure 5 shows that an ensemble calibrated on the present-day mean has improved skill for temperature projections of regional biases compared to the MMM. The reason being that regional biases in temperature trend tend to persist even under different forcing scenarios. We are aware that the persistence of mean bias is not necessarily a demonstration of a constraint on future change. However, the methodology as presented in the manuscript is a proof of concept and has some value even if there are no clear relationships made between mean state bias and globally averaged future change. The novelty here is in the methodology itself and not in the result.

However, we fully agree with the following statement: "While the mean can be correct the response to forcing could be very poorly represented in a model for example." We are actually confirming this with our Figure 6. Figure 6 shows that if we try to predict a change signal (later minus earlier time period), then the calibrated ensemble does not improve skill relative to the multi-model mean of all 20 models (for any of the three diagnostics). So, we agree with the reviewer that the way a model responds to forcings is very important and thus a change signal is hard to constrain.

We now also refer to Knutti et al. (2010) when discussing this figure. They state in their abstract: "Model skill in simulating present-day climate conditions is shown to relate only weakly to the magnitude of predicted change". This is consistent with our findings in Figure 6. We added the following sentences to the last paragraph of section 4.2.1:

*"Across all metrics and variables, the subsets show hardly any RMSE improvement compared to the MMM of the 20 model runs, which is consistent with Sanderson et al. (2017). They found only small changes in projected climate change in the US when weighting models with performance on present day mean climate, and it is consistent with the fact that our field has not managed to significantly reduce uncertainties in both transient (Knutti and Sedláček, 2013) and equilibrium warming (Knutti et al., 2017b).*

*Those findings are also consistent with Knutti et al. (2010), who found that there is only a weak relationship between model skill in simulating present-day climate conditions and the magnitude of predicted change. So, a skillful subset under present-day conditions does not guarantee more confidence in future projections. But even if the uncertainties in future projections are not strongly reduced, there is a clear advantage in reducing biases in the present day climate when using those models to drive impact models, as it reduces the need for complex bias correction methods. Ultimately, when models improve further, and the observed trends get stronger, we would expect that such methods do improve the skill of projections."*

The reviewer also suggests: "*Alternatively this should be presented as a new method which clearly requires more work on the diagnostic side before it is actually be applied to weight models.*"
We are not trying to show a large range of possible diagnostics, as this will be very dependent on the specific application. The aim of the manuscript is simply to introduce this new subset-selection approach based on a simple example where multiple observational products are available and test its out-of-sample skill. Section 4.1 discusses how sensitive the results are to user-decisions, so we feel that we sufficiently discussed the fact that every user needs to make very specific choices depending on the application (and this will change the model weights).

**Selecting A Climate Model Subset To Optimise Key Ensemble Properties - Herger et al. (2017)**

**Report #2**

We thank the reviewer for taking the time to review our manuscript. This document outlines our point-by-point responses to the comments made by the reviewer and the improvements we made to the manuscript (*italicised text in quotation marks*).

General comments

In this paper the authors develop a framework to evaluate the process of selecting a subset of simulations from a large multi-model ensemble (CMIP5). This subject is very important because climate model data users often need to use only a small subset of simulations given limited resources for data processing. However, research groups often face many difficulties when selecting simulations because there is no widely accepted approach to do that, and this highly depends on the climate application. Overall, I found the paper to be insightful and well written, and it should ultimately be considered for publication in ESD after the following minor modifications.

The paper introduces three approaches to select a subset of N simulations from a larger ensemble. These approaches are compared within a common benchmark, that is the equally-weighted ensemble averaging of historical climatology among the selected simulations. The first selection method consists of randomly choosing a N-subset of simulations. This procedure can be repeated several times in order to cover the range of uncertainty associated with a random selection, for instance according to the error of the ensemble mean compared with observed climatology. The second method is the "performance ranking ensemble" in which the N best simulations are selected (according to their individual error in reproducing the observed climatology). The third approach is the "optimal ensemble" by which simulations are selected by minimizing a cost function based on 3 terms: 1) mean square error of the ensemble mean, 2) mean square error of individual members, 3) a measure of model dependence.

An important result in this study is that the ensemble mean of a performance ranking ensemble will perform poorly compared with that of an optimal ensemble, and is comparable with the mean of randomly selected ensembles. This is due to the fact that selecting only the best simulations will leave common biases among them, which will not cancel through the averaging procedure. On the other hand, the optimal ensemble allows to minimize the error of the ensemble mean, discard poorly performing models and maximize the cancellation of errors among simulations. Hence, more independence can be expected between simulations of a same optimal ensemble, while several initial conditions members of a single good model can be part of the same performance ranking ensemble. The paper shows well that the optimal ensemble is the best approach compared with both random and ranking ensemble selections. The way the optimal ensemble is selected is based on a flexible cost function, where weight can be applied to

the existing terms while new terms can be added as well (e.g. maximizing the spread among future projections to reduce overconfidence).

I think this paper allows to get new insights on the impact of selecting a subset of simulations from a large ensemble. However, one weak point is the lack of information about how to use this tool in real life for practitioner and what an "optimal ensemble" actually means in this context. For instance, let us investigate the example of regional climate modelling as given in the manuscript. We first assume that a group can only afford to dynamically downscale 5 GCM simulations with their RCM and that they have defined their own cost function. If they hence use the current tool to optimally select a 5-GCM ensemble to downscale with their RCM, and that a few months later after starting the RCM simulations, they discover that there was a bug in the experiment of GCM #5 and that it should be discarded. It is very likely that GCM #1-4 will no more be an optimal subset of size 4. Similar situation would happen if they realize they can afford producing one more simulation, so they would need to select one more GCM and so the new 6-GCM ensemble will neither be optimal. The concept of an optimal ensemble implies that the selection of one member depends on the other ensemble members. I think this is an important limitation in the applicability of the current method to real-life situations. Moreover, the fact that for each ensemble size there is a ranking of several ensembles is difficult to interpret. It seems to me that for slight differences in RMSE and in the cost function, many other different ensembles are possible. So it should be explained in more details how practitioners should deal with the complexity of coexisting similarly optimal ensembles.

In the example given by the reviewer, the optimal subset for any given ensemble size is indeed dependent on the available simulations and the calibration exercise would have to be repeated if the members in the original ensemble change. We believe that the points raised by the reviewer is applicable for any subset selection approach and thus not specific to the one introduced here, and it is true beyond climate: any carefully selected small subset of people that is representative of a larger group will no longer be optimal when adding, removing or replacing just one person.

The reviewer states: "The concept of an optimal ensemble implies that the selection of one member depends on the other ensemble members. I think this is an important limitation in the applicability of the current method to real-life situations."
We do not regard this as a limitation of the approach especially when trying to address the dependence problem. In order to tell if a simulation is essentially "duplicated" one has to be aware of the other simulations in the ensemble. An approach that is not aware of an ensemble as a whole could just as well be random selection. Any subset selection approach that does not make use of the available information about the original ensemble is most likely not optimal. We therefore do not consider this to limit the applicability of this approach (or indeed any approach to subselection). But we added a couple of sentences to section 4 to make this point explicit:
*"Note, that as the selection of one ensemble member depends on the remaining members in the ensemble, the optimal subset is sensitive to the original set of model runs that we start out with. So, if members are added to or removed from the original ensemble, then the optimal subset is likely going to change. Any subset selection approach that does not*

*make use of the available information about the original ensemble is most likely not optimal."*

Apart from being sensitive to the composition of the original ensemble, the optimal subset also changes depending on the choices made in section 4.1.

The reviewer writes: "Moreover, the fact that for each ensemble size there is a ranking of several ensembles is difficult to interpret."
We assume the reviewer's comment refers to the fact that the second best ensemble of size K might be only slightly worse than the best ("optimal") one, yet have an entirely different makeup. We think that this lack of "smoothness" is inevitable if the goal is to find the single best optimal subset. If ensemble member consistency (with increasing subset size K) is of importance for the user, then he/she would have to be willing to accept something less optimal. Other methods that do maintain this consistency have been proposed (Sanderson et al. (2015), their Fig. 4) if that is desirable, but they are not optimal from an ensemble mean point of view. We now mention this in the manuscript:
*"Another characteristic of the optimal ensemble is that there is not necessarily any ensemble member consistency (with increasing subset size). There are other methods which do maintain this consistency (e.g. Sanderson et al. (2015b)), but such an ensemble is no longer optimal from an ensemble mean point of view."*

However, this is a subjective decision and we therefore do not want to give a general recommendation about how to deal with this issue. Similarly, the black curve in Figure 1 is often relatively flat around the optimal subset, which gives the user the freedom to pick a different subset with slightly worse performance (see second paragraph in section 4). Of course, out-of-sample performance for this particular subset would still have to be ensured.

Another point that should be improved is the last part of the introduction, which doesn't clearly explain the structure of the paper (as there are many subsections) and what is aimed to be achieved. There are also few explanations in the text that are not very clear or lacking some details. See specific comments below.

We addressed this point in our response to "P4L4-34" below.

Specific comments

- P3L19-21 "Regional dynamical downscaling presents a slightly different problem to the one stated above, as the goal is to find a small subset that reproduces certain statistical characteristics of the full ensemble." I know what the authors mean but this paragraph is lacking context about regional climate models, whose goal is to obtain high resolution climate simulations based on lateral boundary conditions taken from GCMs or reanalyses. See for instance:

+ Laprise, R. (2008) Regional Climate Modelling, Journal of Computational Physics, 227(7), 3641–3666. http://dx.doi.org/10.1016/j.jcp.2006.10.024

We thank the reviewer for making us aware of this lack of clarity. We added the reference given and adjusted the sentence to better state the goal of regional downscaling. "*Regional dynamical downscaling presents a slightly different problem to the one stated above, as the goal of regional climate models is to obtain high resolution climate simulations based on lateral boundary conditions taken from global climate models (GCMs) or reanalyses (Laprise et al., 2008). One might therefore attempt to find a small subset of GCMs that reproduces certain statistical characteristics of the full ensemble.*"

- P4L4-34 I think there are too many technical details in the last part of the introduction. As said previously, it should better explain the whole structure of the document. For instance, the three sub-sampling strategies are not explicitly mentioned here. Moreover, the paper contains many subsections, so giving the general plan in the introduction would be useful to the reader.

We agree with the reviewer that as the structure of the manuscript is not traditional, we should motivate better why this structure makes sense. This is how we now describe the structure in the revised manuscript (last paragraph of the introduction):
*"In the following section, we introduce the model data and observational products used throughout this study. Section 3 contains a description of the method used, which includes the pre-processing steps of the data and three ensemble sub-sampling strategies, one of which is the novel approach of finding an optimal ensemble. In section 4 we examine the results by first giving the most basic example of the optimisation problem. We then expand on this example by examining the sensitivity of those results to different choices of the user and highlight the method's flexibility in section 4.1. Out-of-sample skill is tested in section 4.2.1 using model-as-truth experiments to ensure that our approach is not overfitting on the current present-day state. Once that has been ensured, we present future projections based on this novel approach (section 4.2.2). Finally, section 5 contains the discussions and conclusions."*

We decided not to mention the three sub-sampling strategies in the introduction, as this would add even more technical details and it is not essential to introduce the idea.

- P5L24 How did the authors determined that 100 iterations were enough ? Would the error bars change by a lot if one would use 200, or 1000 iterations instead ?

We have redone Figure 1 using 500 rather than 100 iterations and the error bars are not very sensitive to the number of iterations. We added the following sentence to section 3: *"The uncertainty range was found not to be very sensitive to the number of iterations.."*. Plots are shown below.

Surface Temperature, 100 iterations (as in manuscript):

[Figure]

Surface Temperature, 500 iterations:

Total Precipitation, 100 iterations (as in manuscript):

Total Precipitation, 500 iterations:

[Figure]

- P6 What is the reason for the drop between 30-35 members for the performance ranking ensemble ? Could it be due to the fact that some models have several members ?

Here is a list of the first 40 ensemble members ranked by performance:
(1) MPI-ESM-LR_r1i1p1, (2) MPI-ESM-LR_r3i1p1, (3) MPI-ESM-LR_r2i1p1,
(4) MPI-ESM-MR_r1i1p1, (5) CCSM4_r2i1p1, (6) CCSM4_r5i1p1, (7) CCSM4_r1i1p1,
(8) CESM1-BGC_r1i1p1, (9) CCSM4_r3i1p1, (10) CCSM4_r4i1p1, (11) CCSM4_r6i1p1,
(12) ACCESS1-0_r1i1p1, (13) GFDL-CM3_r1i1p1, (14) bcc-csm1-1-m_r1i1p1,
(15) CESM1-CAM5_r3i1p1, (16) CESM1-CAM5_r1i1p1, (17) CESM1-CAM5_r2i1p1,
(18) HadGEM2-AO_r1i1p1, (19) MRI-CGCM3_r1i1p1,  (20) HadGEM2-ES_r4i1p1,
(21) HadGEM2-ES_r3i1p1, (22) HadGEM2-ES_r2i1p1, (23) ACCESS1-3_r1i1p1,
(24) HadGEM2-ES_r1i1p1, (25) MIROC5_r1i1p1, (26) MIROC5_r2i1p1,
(27) IPSL-CM5A-MR_r1i1p1, (28) MIROC5_r3i1p1, (29) CMCC-CMS_r1i1p1,
**(30) GFDL-ESM2M_r1i1p1, (31) GISS-E2-R_r2i1p1, (32) GISS-E2-R-CC_r1i1p1,**
**(33) GISS-E2-R_r1i1p1, (34) CMCC-CM_r1i1p1, (35) GISS-E2-H-CC_r1i1p1,**
(36) CSIRO-Mk3-6-0_r1i1p1, (37) GFDL-ESM2G_r1i1p1, (38) CSIRO-Mk3-6-0_r9i1p1,
(39) CSIRO-Mk3-6-0_r2i1p1, (40) GISS-E2-R_r1i1p2,...

Ensemble size 30-35 are highlighted in bold. It is evident that many GISS-models are part of those ensembles. The GISS models are characterised by errors that are highly uncorrelated with other models, so that adding them to the ensemble tends to cancel out shared biases with the existing 30 member ensemble. This leads to the ensemble mean being closer to observations (and thus the RMSE decreases).

- P6L31 Is there any relationship between the minimum of the optimal ensemble curve (in Fig. 1; between 5 and 8 members for temperature and around 12 for precipitation) and the effective number of independent models in the ensemble ?

There have been several attempts to define what the "effective number of independent models" might mean (e.g., Pennell and Reichler (2011), Jun et al. (2008), Bishop and Abramowitz (2013), Sanderson et al. (2015), Annan and Hargreaves (2011)). However, each of these assume different definitions of independence and have different applications so that there is no clear definition of an effective ensemble size. We added the following sentence to the manuscript:
*"One could investigate defining the effective number of independent models for a given application based on the optimal ensemble size (Annan and Hargreaves, 2011; Bishop*

*and Abramowitz, 2013; Sanderson et al., 2015b; Jun et al., 2008; Pennell and Reichler, 2011), but we have not explored this idea in any detail."*

- P10L17 Why did the authors choose f3 to be the pairwise MSE rather than the pairwise correlation of errors as shown in figure 2 ?

We tried to use error correlation rather than pairwise MSE for term f3, but Gurobi (the mathematical solver we are using) did not converge to a solution (in a reasonable amount of time). This is why we moved to pairwise MSE as an alternative representation of model interdependency.
Note, that the MSE can be decomposed into mean bias, correlation and standard deviation difference, so there is some correlation signal in there (see Gupta et al., Journal of Hydrology, 2009).

- P10L20 "This is a way to address dependence in ensemble spread." Would be worth adding here that it prevents from selecting several members from the same model as well.

We added the following sentence: *"It also makes it harder for the algorithm to select multiple members from the same model."* We decided not to use "prevent", as the other two terms part of the objective function could potentially lead to multiple members from the same model being part of the optimal subset.

- P10L27 "the members of the optimal ensemble", I guess we mean the 3-term one but it is not clear. Also "have a better average performance", it is not clear at all in figure 1 that the 3-term optimal selection is better than the 1-term one, and even the RMSE of the 3-term selection seems a bit higher (triangle are a bit higher than the circles).

The reviewer is right with his/her assumption that we mean the 3-term one. We added *"...based on the cost function with three terms..."* to clarify. When we talk about the better average performance, we refer to the individual members of the optimal ensemble and not the subset averages for each ensemble size. The former is not shown in Figure 1. The fact that the triangles are higher than the circles is mentioned in an earlier sentence (*"Results show that the RMSE of the optimal ensemble mean based on eq. (2) is almost as low as for eq. (1), see Figure 1 (black circles for eq. (1) and triangles for eq. (2))."*)

- P11L19 What do the authors exactly mean by "whether the approach is fitting short term variability" ?

A statistical model that is fitting short term variability is essentially overfitting on the available data. Skill in-sample (where observations are available) does not guarantee skill out-of-sample (without observations), which is the motivation for the model-as-truth experiment. We added the following sentence to clarify: "*In other words, we have to ensure that our subset-selection approach is not overfitting on the available data in-sample which can potentially lead to spurious results out-of-sample*".

- P12L1-2 The different metrics (trend and space+time) should be defined more explicitly.

We agree with the reviewer that the way we introduce those diagnostics was not very clear. In this sentence, we are now referring to Section 4.1 (paragraph: "Alternatives to climatology"), where those metrics are introduced. We adjusted the paragraph entitled "Alternatives to climatology". It now reads as follows:

*"As part of the data pre-processing step, we computed climatologies (time-means at each grid cell) for the model output and observational dataset. In addition to climatologies, we decided to consider time-varying diagnostics ("trend" and "space+time"), which potentially contain information relevant for projections, which is not captured by time-means. For the "trend" diagnostic, we compute a linear trend of the corresponding variable at each grid cell and end up with a two dimensional array for each simulation and observational dataset. As a second time-varying diagnostic, we compute 10-year running means at each grid-cell to obtain a three dimensional array which is subsequently used for the analysis. This is hereafter referred to as "space+time". Subsection 4.2.1 shows (based on a model-as-truth experiment) how sensitive the ensemble can be to the diagnostics (mean, trend, or variability) chosen in the pre-processing step."*

- P12L14 "all available runs" Do we mean 81 runs or one per institute ? Please clarify here and elsewhere in the text.

All model-as-truth experiments in Section 4.2 are conducted with one run per modelling institution (21 simulations). So, "all available runs" refers to 20 runs (as 1 of the runs is the "truth"). We have now clarified this here and elsewhere in the text.
*"It is evident that both the climatological metric and the space + time metric have improved skill out-of-sample compared to simply taking the mean of all 20 runs."*

- P13L31 "mean of all 81 model runs" As the authors use this example very often in the paper, it should I think be stated somewhere that this is really bad practice to average all models and realizations in a CMIP experiment because we arbitrarily give more weight to the models represented by the largest number of members. It should also be made more clear why they use this benchmark rather than simply the multi-model ensemble mean based on one realization per model (that is 38 models) ?

Figure 1 contains horizontal lines, which are based on different ways to calculate a multi-model mean. The dashed horizontal line in Figure 1 shows results of the mean of 38 models. However, different to what the reviewer suggested the 81 runs are first averaged across initial condition members before the mean of the 38 models is taken. We don't expect the results to change much if instead one member per model was chosen. We added a sentence stating that averaging across all models and realizations in CMIP is bad practice ("Defining the benchmark" paragraph in Section 4.1): *"We would consider this bad practice as we arbitrarily give more weight to the models represented by the largest number of members."*
Throughout the model-as-truth experiment (Figures 4-6) one run per institution was chosen as our benchmark (we therefore no longer have multiple runs per model).
The use of the MMM of 81 model runs as a benchmark is therefore limited to the maps in Figures 7 and 8. We agree with the reviewer that this choice of benchmark is bad practice and have thus redone those maps. As a benchmark, we now average across initial

condition members before averaging across the 38 models. We have adjusted the text accordingly. The maps are very similar to the ones based on the mean of 81 model runs.

- P15L3-4 Similarly to the issue of a selection based on multiple variables and observational dataset (as pointed in the previous reviews), many applications such as impacts assessments or regional climate modelling should imply an ensemble selection based on a specific region. So the fact that the optimal ensemble will depends of the region where it is calibrated should be discussed as well.

This is a good point! This aspect has been addressed to a certain extent given that the observational products we used vary in their spatial coverage. We added the following sentences about the sensitivity of the optimal subset to the chosen spatial domain in Section 4.1 ("Choice of observational product"):

*"HadCRUT4 for example is the same as CRUTEM4 over land, but additionally has data over the ocean. The optimal subsets derived from calibrating on those two observational products separately are quite different, which highlights the sensitivity of the calibration exercise to the chosen spatial domain. This is particularly important for impact assessments and regional climate modelling, where ensemble selection is done based on a specific region."*

- Figure 2: The label of the y-axis is misleading because a high error correlation rather implies model-model similarity.

This is correct. We have changed the y-axis to "Dependence Error Correlation: Model-Model Similarity" and also adjusted the title of this figure.

- Figure 3: Regional downscaling should be as well in the "Application to the future", not only in the "historical data" blue box.

We agree with the reviewer and have adjusted Figure 3 accordingly.

[revised manuscript text omitted]